# STACEY: Promoting Stochastic Steepest Descent via Accelerated $\ell_p$-Smooth Nonconvex Optimization

Xinyu Luo [* 1]   Cedar Site Bai [* 1]   Bolian Li [* 1]   Petros Drineas [1]   Ruqi Zhang [1]   Brian Bullins [1]

## Abstract

While popular optimization methods such as SGD, AdamW, and Lion depend on steepest descent updates in either $\ell_2$ or $\ell_\infty$ norms, there remains a critical gap in handling the non-Euclidean structure observed in modern deep networks training. In this work, we address this need by introducing a new *accelerated* $\ell_p$ steepest descent algorithm, called STACEY, which uses interpolated primal-dual iterate sequences to effectively navigate non-Euclidean smooth optimization tasks. In addition to providing novel theoretical guarantees for the foundations of our algorithm, we empirically compare our approach against these popular methods on tasks including image classification and language model (LLM) pretraining, demonstrating both faster convergence and higher final accuracy. We further evaluate different values of $p$ across various models and datasets, underscoring the importance and efficiency of non-Euclidean approaches over standard Euclidean methods. Code can be found at https://github.com/xinyuluo8561/Stacey.

## 1. Introduction

Stochastic first-order methods have proven essential for efficiently training modern deep learning models. Beyond the basic stochastic gradient descent (SGD) algorithm (Robbins & Monro, 1951) and its momentum-based variants (Nesterov, 1983; Polyak, 1964), a variety of adaptive methods have been developed, such as AdaGrad (Duchi et al., 2011a), Adam (Kingma & Ba, 2015), and AdamW (Loshchilov & Hutter, 2019), which incorporate second-moment gradient information to provide per-coordinate scaling. Meanwhile,

more recent methods like signSGD (Bernstein et al., 2018) and Lion (Chen et al., 2023) focus on using the *sign* of the (stochastic) gradient.

Although these algorithms have shown impressive empirical performance (sometimes exceeding that of standard adaptive methods), their theoretical analyses typically rely on Euclidean (i.e., $\ell_2$) or $\ell_\infty$-based assumptions. Specifically, crucial to guarantees of finding $\epsilon$-approximate stationary points (Carmon et al., 2017; Ghadimi & Lan, 2013; Jin et al., 2017) are two related choices: (i) the norm used to define stationarity, and (ii) the corresponding notion of smoothness. Classical analyses in deep learning often adopt Euclidean smoothness (Ghadimi & Lan, 2013), while signSGD relies on $\ell_\infty$-based assumptions (Bernstein et al., 2018; Balles et al., 2020).

Yet, there is mounting evidence—both theoretical and empirical—suggesting that a more flexible $\ell_p$ perspective can capture the geometric structure of complex deep network objectives far better than *either* $p = 2$ or $p = \infty$ alone (Adolphs et al., 2019; Cohen et al., 2021; Ghorbani et al., 2019; Jiang et al., 2024; Li et al., 2020a; Papyan, 2018). For instance, depending on the shape of the loss surface and the distribution of gradients across coordinates, certain $\ell_p$ norms with $2 < p < \infty$ may lead to faster descent or improved generalization. This leaves open a significant gap: How can we develop and analyze optimizers in *alternative* non-Euclidean regimes, namely those with *general* $\ell_p$ norms where $p \in (2, \infty)$?

To address this question, we propose a novel approach that we term STACEY (**St**ochastic **St**eepest Descent with **Acce**leration). Our development builds on insights from both $\ell_p$-steepest descent and non-Euclidean acceleration techniques (Allen-Zhu & Orecchia, 2017; Diakonikolas & Guzmán, 2024; Nemirovskii & Nesterov, 1985; Nesterov, 2005), combining primal-dual (Diakonikolas & Orecchia, 2019) iterates with an interpolation scheme specifically for $\ell_p$-based smoothness. While the notion of acceleration is well understood in the classical (Euclidean) setting (Nesterov, 1983), extending it to arbitrary $\ell_p$ norms introduces a fundamental trade-off: although we may attain improved geometry dependence (and thus potentially faster practical convergence in certain regimes), the theoret-

---

[*]Equal contribution  [1]Department of Computer Science, Purdue University, Indiana, USA. Correspondence to: Xinyu Luo <luo466@purdue.edu>, Cedar Site Bai <bai123@purdue.edu>, Bolian Li <li4468@purdue.edu>.

*Proceedings of the 42nd International Conference on Machine Learning*, Vancouver, Canada. PMLR 267, 2025. Copyright 2025 by the author(s).

ical "acceleration exponent" necessarily decreases from 2 toward 1 as $p$ grows large (Guzmán & Nemirovski, 2015; Nemirovskii & Nesterov, 1985). Nonetheless, by situating STACEY within this continuum of non-Euclidean optimizers, we can reap meaningful benefits over purely Euclidean (e.g., SGD) and purely sign-based (e.g., signSGD) methods on modern, large-scale tasks.

**Our Contributions:**

- **Accelerated $\ell_p$-based method.** Drawing inspiration from primal-dual interpolation techniques in the convex setting (Allen-Zhu & Orecchia, 2017; Nesterov, 2005; Diakonikolas & Guzmán, 2024), we design STACEY, an *accelerated $\ell_p$* descent algorithm specifically tailored to non-Euclidean smooth optimization (Section 4).

- **General $\ell_p$ convergence guarantees for non-convex problems.** We first establish *stochastic $\ell_p$ steepest descent* guarantees $\mathbb{E}\left[\|\nabla f(\hat{x})\|_{p^*}^{p^*}\right] \leq \epsilon$ at a rate of $O(\epsilon^{-4})$, under standard variance-bounded and $\ell_p$-smoothness assumptions, where we let $p^* := \frac{p}{p-1}$ (Section 4.1). Our results strictly generalize previous guarantees for signSGD ($p = \infty$).

- **Practical performance on large-scale tasks.** We compare STACEY against popular optimizers such as SGD, Adam, AdamW, and Lion on tasks ranging from image classification to pretraining large language models (Section 5). Our experiments show that STACEY can converge faster and achieve higher accuracy than these baselines, particularly when the geometry of the objective departs significantly from the Euclidean setting.

- **Flexible norm choices.** We further evaluate different values of $p \in (2, \infty)$ across various model architectures and datasets, illustrating the potential advantages of tailoring the choice of norm to the problem geometry.

Taken together, our results highlight the importance of *non-Euclidean* perspectives for contemporary machine learning tasks, offering both theoretical insight and practical improvement over classical ($\ell_2$-based) and sign-based ($\ell_\infty$-based) optimizers.

## 2. Related Work

**Methods for non-Euclidean geometries.** A significant line of research has studied *sign-based* methods, which can be viewed as (stochastic) steepest descent under the $\ell_\infty$ norm. For instance, Bernstein et al. (2018) introduced signSGD and analyzed its convergence properties through

an $\ell_2$ majorization-based smoothness condition,[1] showing that in expectation, $\|\nabla f(\hat{x})\|_1$ can be driven below a prescribed threshold. Similarly, Balles et al. (2020) investigated the geometric underpinnings of sign-based updates, highlighting how they relate to $\ell_\infty$ steepest descent. Recent work on Lion (Chen et al., 2023) and its generalization Lion-$\mathcal{K}$ (Chen et al., 2024) further underscores the empirical benefits of sign-driven coordinates in large-scale tasks.

However, sign-based approaches ($p = \infty$) represent just one extreme of non-Euclidean geometry. The other well-studied example is the classical $\ell_2$-based regime (e.g., vanilla SGD) (Ghadimi & Lan, 2013; Robbins & Monro, 1951), where standard notions of Euclidean smoothness and approximate stationarity $\|\nabla f(\hat{x})\|_2 \leq \epsilon$ underpin core theoretical results. Interpolating between these extremes ($\ell_2$ and $\ell_\infty$) by considering $\ell_p$ norms for $2 < p < \infty$ has remained comparatively underexplored in the stochastic, non-convex setting. One challenge is that, unlike in the Euclidean case, the coordinate-scaling in an $\ell_p$ steepest-descent update is not merely a straightforward unbiased estimator of the full-batch direction, making a standard "SGD-style" analysis more involved.

**Methods for curvature-aware optimization.** Another line of work exploits local geometry by incorporating second-order information, such as the Hessian or Fisher information matrix, and develops techniques for their efficient approximation. K-FAC (Martens & Grosse, 2015) approximates the Fisher information matrix using layer-wise Kronecker-factored preconditioners for efficient second-order updates. Shampoo (Gupta et al., 2018; Morwani et al., 2025; Vyas et al., 2025) similarly employs per-dimension Kronecker-factored preconditioners to approximate the gradient's second-moment matrix, enabling scalable curvature-aware optimization for tensor-structured parameters. Sophia (Liu et al., 2024) further improves scalability by approximating the diagonal of the Hessian using second-order momentum. In contrast, our method, STACEY, is a first-order approach that leverages non-Euclidean geometry, rather than local curvature, through a differing $\ell_p$ norm.

**Why $\ell_p$-based methods help for large models.** A key motivation for exploring $\ell_p$-norms with $p \in (2, \infty)$ stems from recent studies on the Hessian spectrum of large neural networks (Ghorbani et al., 2019; Papyan, 2018). In particular, Ghorbani et al. (2019) provide evidence that the Hessian eigenvalue density can be highly non-uniform, leading to large curvature in certain subspaces while others remain comparatively flat. Under standard $\ell_2$-based (Euclidean) assumptions, these directions of high curvature can inflate the global smoothness parameter $L_2$, potentially slowing con-

---

[1]We provide a comparison of $\ell_2$ majorization and $\ell_p$ smoothness conditions in Appendix B.

vergence or complicating optimization. By transitioning to $\ell_p$-smoothness for $p > 2$, one can sometimes leverage a reasonable Lipschitz constant $L_p$, as high-curvature directions may not always dominate in the same way.

Formally, a function $f$ is $\ell_p$-smooth if, for all $x, y \in \mathbb{R}^d$,

$$\|\nabla f(x) - \nabla f(y)\|_{p^*} \leq L_p \|x - y\|_p,$$

where $p^* = \frac{p}{p-1}$ is the dual value. Thus, the choice of $p$ shifts how curvature in different coordinates or subspaces affects $\nabla f$. Because large-scale models often exhibit anisotropic Hessians (Cohen et al., 2021; Li et al., 2020a), an $\ell_p$ analysis can better mirror the true geometric structure of the objective. This observation aligns with analyses in (Balles et al., 2020), where sign-based methods (i.e., $\ell_\infty$) can exploit flat directions effectively; by continuity, $\ell_p$ norms for $p \in (2, \infty)$ may interpolate between purely Euclidean and purely sign-driven behaviors.

Two main factors motivate the study of general $\ell_p$-norms ($2 < p < \infty$) in large-scale training:

1. **Hessian Geometry and Tail Behavior.** Large neural networks often exhibit Hessians whose eigenvalues and singular vectors follow nontrivial (sometimes heavy-tailed) distributions (Ghorbani et al., 2019; Papyan, 2018). By choosing $p$ to better accommodate outlier directions or to exploit more uniform curvature across coordinates, one can leverage better effective $\ell_p$-smoothness constant $L_p$.

2. **Balancing Sparse and Dense Updates.** Methods at $p = \infty$ (sign-based) produce coordinate-wise updates of the same magnitude, while $\ell_2$-based approaches "spread out" updates proportionally to gradient magnitudes. In high dimensions, intermediate $\ell_p$ steps can yield a better trade-off between these extremes, potentially improving both speed of descent and generalization (Cohen et al., 2021; Li et al., 2020a).

**Trade-offs for non-Euclidean acceleration.** Alongside this matter of defining (and parameterizing) smoothness, there is a second lens through which we observe the potential for general $p$, *namely that of acceleration* (Allen-Zhu & Orecchia, 2017; Bai & Bullins, 2024a; Nemirovskii & Nesterov, 1985; Nesterov, 1983; 2005). As we further discuss in Section 4, there is a fundamental trade-off (for convex settings) between the rate of acceleration and the norm used to measure the initial distance to the optimal solution. Concretely, it is well known that, for convex $f(x)$ that is $L$-smooth with respect to $\|\cdot\|_2$, the classic accelerated gradient descent (AGD) method of (Nesterov, 1983) converges at the rate $f(x_T) - f(x^*) \leq O\left(\frac{L\|x_0 - x^*\|_2^2}{T^2}\right)$, and this rate is indeed tight (Nesterov, 2018; Nemirovskij & Yudin, 1983).

Importantly, we emphasize the appearance here of $\|\cdot\|_2$ not only in measuring smoothness, but also for the $\|x_0 - x^*\|_2^2$ term.

Unfortunately, the standard analysis of AGD does not readily adapt to alternative notions of smoothness, as the design of the algorithm is, in a sense, *specific to Euclidean settings*; we refer the reader to the work of (Allen-Zhu & Orecchia, 2017) for further discussion of this basic incompatibility. Nevertheless, several works (Diakonikolas & Guzmán, 2024; Nemirovskii & Nesterov, 1985; Nesterov, 2005; Song et al., 2021)—including that of Allen-Zhu & Orecchia (2017)—with optimal rates in the Euclidean setting (Nesterov, 2018; Bai & Bullins, 2025), have provided techniques for *accelerating in non-Euclidean settings*. In particular, the approach of Nemirovskii & Nesterov (1985), for convex $f(x)$ that is $L_p$-smooth with respect to $\|\cdot\|_p$, leads to guarantees of the form

$$f(x_T) - f(x^*) \leq O\left(\frac{L_p\|x_0 - x^*\|_p^2}{T^{\frac{p+2}{p}}}\right). \qquad (1)$$

(See also, e.g., Theorem 2 in (Diakonikolas & Guzmán, 2024).) Moreover, these rates are likewise known to be tight (Guzmán & Nemirovski, 2015).

Looking closely at these convergence guarantees, we may first note that, for $p = 2$, the rate in equation 1 recovers that of Nesterov (1983). On the other hand, for $p \to \infty$, while $\|x_0 - x^*\|_p^2$ can, at best, be as small as $d^{\frac{2}{p}-1}\|x_0 - x^*\|_2^2$, we also have that $\lim_{p \to \infty} T^{-\frac{p+2}{p}} = T^{-1}$—*in which case the benefit of acceleration disappears altogether*—and in fact this (limiting) rate essentially matches that of *unaccelerated* $\ell_\infty$ steepest descent (Kelner et al., 2014). Consequently, these observations reveal the opportunity afforded by (non-Euclidean) $\ell_p$-based accelerated methods **for other values of** $p < \infty$, resulting from this trade-off between the *dependence on the problem geometry* and the *rate of acceleration*.

## 3. Preliminaries and Assumptions

Throughout we let $\|\cdot\|$ and $\|\cdot\|_*$ denote a general norm and its dual, respectively. In addition, we specify $\|\cdot\|_p$ to denote the standard $\ell_p$ norm ($1 \leq p \leq \infty$) and $\|\cdot\|_{p^*} := \|\cdot\|_{p/(p-1)}$ to denote its dual norm. For symmetric $M \in \mathbb{R}^{d \times d}$ s.t. $M \succ 0$, we further let $\|\cdot\|_M$ denote the standard matrix norm, i.e., $\|x\|_M = \sqrt{x^\top M x}$ for $x \in \mathbb{R}^d$. For a vector $v \in \mathbb{R}^d$, we use superscript, i.e., $v^{(i)}$ to denote the $i^{th}$ coordinate of $v$, and we let $\text{diag}(v)$ denote the diagonal matrix such that $\text{diag}(v)_{i,i} = v^{(i)}$. We use subscript, e.g., $\theta_t$, to denote a vector in the $t^{th}$ iteration. For brevity, we use $g_t$ for the true gradient $\nabla f(\theta_t)$ and $\tilde{g}_t$ for the stochastic gradient $g(\theta_t)$. We use $\text{sgn}(\cdot)$ to denote the sign function and $\mathbb{I}_{[\cdot]}$ to denote the indicator function.

**Algorithm 1** STACEY$_{(p,2)}$ Optimizer

**input** $p, \beta_1, \beta_2, \alpha, \tau, \eta, \epsilon, \lambda, f$
**initialize** $\theta_0, z_0, m_0 \leftarrow 0$

1: **while** $\theta_{t+1}$ not converged **do**
2: $\quad \tilde{g}_t \leftarrow \tilde{g}$ s.t. $\mathbb{E}[\tilde{g}] = \nabla f(\theta_t)$
3: $\quad c_{t+1} \leftarrow \beta_1 m_t + (1 - \beta_1)\tilde{g}_t$
4: $\quad s^\epsilon(x) = [s_1^\epsilon(x), \cdots, s_d^\epsilon(x)]^\top$ where

$$s_i^\epsilon(x) = \frac{x^{(i)}}{\left|x^{(i)}\right|^{\frac{p-2}{p-1}} + \epsilon}, \ \forall i \in [d]$$

5: $\quad y_{t+1} \leftarrow \theta_t - \eta_t s_\epsilon(c_{t+1})$
6: $\quad z_{t+1} = z_t - \alpha c_{t+1}$
7: $\quad \theta_{t+1} = \tau z_{t+1} + (1 - \tau)y_{t+1} - \eta_t \lambda \theta_t$
8: $\quad m_{t+1} = \beta_2 m_t + (1 - \beta_2)\tilde{g}_t$
9: **end while**
10: **return** $\theta_{t+1}$

---

**Algorithm 2** STACEY$_{(p,p)}$ Optimizer

**input** $p, \beta_1, \beta_2, \alpha, \tau, \eta, \epsilon, \lambda, f$
**initialize** $\theta_0, z_0, m_0 \leftarrow 0$

1: **while** $\theta_{t+1}$ not converged **do**
2: $\quad \tilde{g}_t \leftarrow \tilde{g}$ s.t. $\mathbb{E}[\tilde{g}] = \nabla f(\theta_t)$
3: $\quad c_{t+1} \leftarrow \beta_1 m_t + (1 - \beta_1)\tilde{g}_t$
4: $\quad s^\epsilon(x) = [s_1^\epsilon(x), \cdots, s_d^\epsilon(x)]^\top$ where

$$s_i^\epsilon(x) = \frac{x^{(i)}}{\left|x^{(i)}\right|^{\frac{p-2}{p-1}} + \epsilon}, \ \forall i \in [d]$$

5: $\quad y_{t+1} \leftarrow \theta_t - \eta_t s^\epsilon(c_{t+1})$
6: $\quad z_{t+1}^{(i)} = \frac{\left|z_t^{(i)}\right|^{p-2} z_t^{(i)} - \alpha c_{t+1}^{(i)}}{\left|\left|z_t^{(i)}\right|^{p-2} z_t^{(i)} - \alpha c_{t+1}^{(i)}\right|^{\frac{p-2}{p-1}} + \epsilon}, \forall i \in [d]$
7: $\quad \theta_{t+1} = \tau z_{t+1} + (1 - \tau)y_{t+1} - \eta_t \lambda \theta_t$
8: $\quad m_{t+1} = \beta_2 m_t + (1 - \beta_2)\tilde{g}_t$
9: **end while**
10: **return** $\theta_{t+1}$

---

We may also consider the following equivalent definition of $\ell_p$ smoothness.

**Assumption 1** (Smoothness in $\ell_p$ norm). *Let $f : \mathbb{R}^d \mapsto \mathbb{R}$ be L-smooth w.r.t. $\|\cdot\|_p$ for $p \geq 2$. Then, for all $x, y \in \mathbb{R}^d$,*

$$\left|f(y) - f(x) - \nabla f(x)^\top(y - x)\right| \leq \frac{L}{2}\|y - x\|_p^2.$$

## 4. Accelerating Stochastic Steepest Descent

Inspired by previous techniques in non-Euclidean acceleration (Allen-Zhu & Orecchia, 2017; Nesterov, 2005)—as well as their successes, e.g., (Bullins, 2020; Jambulapati et al., 2019; Sherman, 2017; Sidford & Tian, 2018)—we introduce a practical acceleration scheme called STACEY (Algorithm 1), which is *specifically designed for $\ell_p$-based methods*. Central to our approach is its reliance on *two* sequences of stochastic steps: 1) one sequence based on the standard $\ell_p$ steepest descent direction (line 5), which we show is theoretically well-grounded in the stochastic nonconvex setting; 2) another sequence—whose combination with the first ultimately leads to acceleration—based on a gradient descent direction (line 6), whose details we will further discuss.

### 4.1. Analyzing Stochastic $\ell_p$ Descent

In this section, we present the stochastic $\ell_p$ descent algorithm, which serves as the fundamental framework of our approach, and establish its convergence guarantees. As shown in Algorithm 3, its update step takes the unscaled form[2] of its counterpart in the deterministic setting

---

**Algorithm 3** Stochastic $\ell_p$ Descent

**input** $p, \eta, f, \theta_0$

1: **for** $t = 0$ **to** $T - 1$ **do**
2: $\quad s(x) = [s_1(x), \cdots, s_d(x)]^\top$ where

$$s_i(x) = \frac{x^{(i)}}{\left|x^{(i)}\right|^{\frac{p-2}{p-1}}}, \ \forall i \in [d]$$

3: $\quad \theta_{t+1} = \theta_t - \eta s(\tilde{g}_t) \qquad \triangleright \tilde{g}_t$ s.t. $\mathbb{E}[\tilde{g}_t] = \nabla f(\theta_t)$
4: **end for**
5: **return** $\theta_T$

---

$\theta_{t+1}^{(i)} = \theta_t^{(i)} - \eta\|g_t\|_{p^*}^{\frac{p-2}{p-1}} \frac{g_t^{(i)}}{\left|g_t^{(i)}\right|^{\frac{p-2}{p-1}}}$, which is derived from the closed form of

$$\theta_{t+1} = \arg\min_\theta \left\{ \langle \eta g_t, \theta - \theta_t \rangle + \frac{1}{2}\|\theta - \theta_t\|_p^2 \right\}.$$

When $p = \infty$, Algorithm 3 reduces exactly to signSGD (Bernstein et al., 2018).

For $p > 2$, we show in Theorem 1 that stochastic $\ell_p$ descent converges in expectation to an $\epsilon$-approximate stationary point with respect to the dual norm at a rate of $O(\epsilon^{-4})$, thereby generalizing the previous guarantees for signSGD ($p = \infty$). In addition, we provide here a proof sketch, deferring the complete proof to Appendix A.1. Curiously, as we will see, moving from the $\ell_2$ setting (or even from the $\ell_\infty$ setting) introduces certain technical considerations that need to be addressed non-trivially. As standard in stochastic and

---

[2]This is in line with signSGD (Bernstein et al., 2018) compared to the scaled form in (Balles et al., 2020). In addition, we adopt the unscaled version for clearer convergence analysis and a more practical implementation.

non-Euclidean settings (Ghadimi & Lan, 2013; Bernstein et al., 2018), we rely on the following assumptions.

**Assumption 2** (Unbiased Estimate). *The stochastic gradient $\tilde{g}$ is an unbiased estimate of the true gradient $g$. That is, $\mathbb{E}[\tilde{g}] = g$.*

**Assumption 3** (Bounded Variance). *For some data $\xi$, the variance of each coordinate of the stochastic gradient is bounded, i.e., $\forall i \in [d]$, $\mathbb{E}[|\tilde{g}^{(i)} - g^{(i)}|^2] \leq \sigma_i^2$.*

**Corollary 1.** *By Assumption 3, $\mathbb{E}[\|\tilde{g} - g\|_2^2] \leq \sigma^2$ where for $\sigma := \|\vec{\sigma}\|_2$, $\vec{\sigma} = [\sigma_1, \cdots, \sigma_d]^\top$.*

**Corollary 2.** *If the stochastic gradient is an $n$-sample minibatch estimate, then $\forall i \in [d]$, $\mathbb{E}[|\tilde{g}^{(i)} - g^{(i)}|^2] \leq \frac{\sigma_i^2}{n}$.*

**Assumption 4** (Bounded gradient). *For $G > 0$, $p \geq 2$, and $p^*$ where $\frac{1}{p} + \frac{1}{p^*} = 1$, $\|\tilde{g}\|_{p^*} \leq G$.*

**Corollary 3.** *By Assumption 4, we know that*

(a) *$\|g\|_{p^*} = \|\mathbb{E}[\tilde{g}]\|_{p^*} \leq \mathbb{E}[\|\tilde{g}\|_{p^*}] \leq G$ with Jensen's inequality.*

(b) *$\forall\, i \in [d]$, $\left|\tilde{g}^{(i)}\right| \leq G$ and $\left|g^{(i)}\right| \leq G$.*

We briefly justify the necessity of Assumption 4, which arises from additional technical challenges. Specifically, the coordinate-wise re-scaled update introduces bias under standard assumptions, preventing the direct application of conventional expectation and variance analyses as we later elaborate in detail. Notably, similar assumptions are also made when analyzing problems with complex structures, such as stochastic compositional (Wang et al., 2017), composite (Wang et al., 2024; Duchi et al., 2011b), and federated optimization (Li et al., 2020b; Yuan et al., 2021; Bai & Bullins, 2024b). Now we introduce the convergence result for $\ell_p$ steepest descent in the stochastic non-convex setting.

**Theorem 1** (Main). *Running Algorithm 3 on some (possibly non-convex) function $f$ that satisfies Assumptions 1 to 4 yields*

$$\mathbb{E}\left[\frac{1}{T}\sum_{t=0}^{T-1}\|g_t\|_{p^*}^{p^*}\right] \leq \frac{f_0 - f^*}{\eta T} + \frac{L\eta G^{\frac{2}{p-1}}}{2}$$
$$+ \frac{1}{T}\sum_{t=0}^{T-1}\frac{\frac{2p-1}{p-1}G^{\frac{1}{p-1}}\|\vec{\sigma}\|_1}{\sqrt{n_t}}$$

*where $f_0 = f(\theta_0)$ and $f^* = f(\theta^*)$, $n_t$ is the batch size in iteration $t$ and $L$, $\vec{\sigma}$, and $G$ are constants from Assumption 1, 3, 4. Further letting the batch size $n_t = T$, the number of gradient call is $N = T^2$ for $T$ iterations. With $\eta =$*

$\frac{1}{L^{\frac{1}{2}}G^{\frac{1}{p-1}}T^{\frac{1}{2}}}$ *we have*

$$\mathbb{E}\left[\frac{1}{T}\sum_{t=0}^{T-1}\|g_t\|_{p^*}^{p^*}\right] \leq$$
$$\frac{1}{N^{\frac{1}{4}}}\left[L^{\frac{1}{2}}G^{\frac{1}{p-1}}\left(f_0 - f^* + \frac{1}{2}\right) + \frac{2p-1}{p-1}G^{\frac{1}{p-1}}\|\vec{\sigma}\|_1\right],$$

*i.e., Algorithm 3 takes $N \in \mathcal{O}\left(\epsilon^{-4}\right)$ gradient queries to reach an $\epsilon$-approximate stationary point.*

*Proof Sketch.* Starting with Assumption 1 and the descent step in Algorithm 3,

$$f(\theta_{t+1}) \leq f(\theta_t) - \underbrace{\eta\langle g_t, s(g_t)\rangle}_{A} + \underbrace{\eta\langle g_t, s(g_t) - s(\tilde{g}_t)\rangle}_{B}$$
$$+ \underbrace{\frac{L\eta^2}{2}\|s(\tilde{g}_t)\|_p^2}_{C},$$

where $A = \eta\|g_t\|_{p^*}^{p^*}$. In conventional first-order analysis, the inner product term $B$ is supposed to cancel out after taking expectation. In contrast, the closed-form stochastic $\ell_p$ descent update is coordinate-wise re-scaled, which makes the descent step *biased*, that is, $\mathbb{E}[s(\tilde{g})] \neq s(f(x))$. In the literature on biased gradient descent (Stich & Ajalloeian, 2020; Demidovich et al., 2023), the bias terms simply accumulate as constants and do not decay with the iterations. Thus, this term requires novel techniques to guarantee convergence. Noticing that $s_i(x) = \frac{x^{(i)}}{|x^{(i)}|^{\frac{p-2}{p-1}}} = \mathrm{sgn}(x^{(i)})|x^{(i)}|^{\frac{1}{p-1}}$,

$$B = \eta\sum_{i=1}^d g_t^{(i)}\left(\mathrm{sgn}\left(g_t^{(i)}\right)|g_t^{(i)}|^{\frac{1}{p-1}} - \mathrm{sgn}\left(\tilde{g}_t^{(i)}\right)|\tilde{g}_t^{(i)}|^{\frac{1}{p-1}}\right)$$
$$= \eta\sum_{i=1}^d\left|g_t^{(i)}\right|\left(|g_t^{(i)}|^{\frac{1}{p-1}} + |\tilde{g}_t^{(i)}|^{\frac{1}{p-1}}\right)\mathbb{I}_{\left[\mathrm{sgn}\left(g_t^{(i)}\right)\neq\mathrm{sgn}\left(\tilde{g}_t^{(i)}\right)\right]}$$
$$+ \eta\sum_{i=1}^d\left|g_t^{(i)}\right|\left||g_t^{(i)}|^{\frac{1}{p-1}} - |\tilde{g}_t^{(i)}|^{\frac{1}{p-1}}\right|\mathbb{I}_{\left[\mathrm{sgn}\left(g_t^{(i)}\right)=\mathrm{sgn}\left(\tilde{g}_t^{(i)}\right)\right]}.$$

Denote the first term as $B_1$ and the second $B_2$. The $|g_t^{(i)}|^{\frac{1}{p-1}} + |\tilde{g}_t^{(i)}|^{\frac{1}{p-1}}$ term in $B_1$ can be bounded by $2G^{\frac{1}{p-1}}$ with Corollary 3, after which we take expectation, turning the indicator into a probability, and Lemma 2 in Appendix A.1 shows $\mathbb{E}[B_1] \leq \frac{2\eta G^{\frac{1}{p-1}}\|\vec{\sigma}\|_1}{\sqrt{n_t}}$ using Markov's inequality.

$B_2$ requires more sophisticated handling since we cannot push the expectation through due to the data dependence of the term $\left||g_t^{(i)}|^{\frac{1}{p-1}} - |\tilde{g}_t^{(i)}|^{\frac{1}{p-1}}\right|$, nor does $\mathbb{P}\left[\mathrm{sgn}\left(g_t^{(i)}\right) = \mathrm{sgn}\left(\tilde{g}_t^{(i)}\right)\right]$ give us much information. We instead take the zeroth-order Taylor expansion so

that $\forall\ i\ \in\ [d],\ \exists\ \zeta^{(i)}$ between $g_t^{(i)}$ and $\tilde{g}_t^{(i)}$ such that

$$|g_t^{(i)}|^{\frac{1}{p-1}} = |\tilde{g}_t^{(i)}|^{\frac{1}{p-1}} + \frac{1}{p-1}\mathrm{sgn}(\zeta^{(i)})\left|\zeta^{(i)}\right|^{\frac{2-p}{p-1}}\left(g_t^{(i)} - \tilde{g}_t^{(i)}\right).$$

In addition, we have

$$\left||g_t^{(i)}|^{\frac{1}{p-1}} - |\tilde{g}_t^{(i)}|^{\frac{1}{p-1}}\right|$$
$$= \frac{1}{p-1}\mathrm{sgn}(\zeta^{(i)})\left|\zeta^{(i)}\right|^{\frac{2-p}{p-1}}\left(g_t^{(i)} - \tilde{g}_t^{(i)}\right).$$

Furthermore, given $\mathrm{sgn}\left(g_t^{(i)}\right) = \mathrm{sgn}\left(\tilde{g}_t^{(i)}\right)$, it is either $\left|g_t^{(i)}\right| \leq \left|\zeta^{(i)}\right| \leq \left|\tilde{g}_t^{(i)}\right|$ or $\left|g_t^{(i)}\right| \geq \left|\zeta^{(i)}\right| \geq \left|\tilde{g}_t^{(i)}\right|$. Appendix A.1 Lemma 3 shows that $\mathbb{E}\left[B_2\right] \leq \frac{\eta G^{\frac{1}{p-1}}\|\vec{\sigma}\|_1}{(p-1)\sqrt{n_t}}$ in either case.

Term $C$ is usually turned into mean-squared error that coincides with variance in an unbiased setting, which the bounded variance assumption can directly handle. This is not the case for our setting. It is worth noting that the analysis of signSGD (Bernstein et al., 2018), a special case of the $\ell_p$ setting with $p = \infty$, was able to push through due to its update being in the very form of the sign of the gradient, which is in itself bounded by the constant 1. Our update, in contrast, is more complicated with the absolute value of the coordinates of the gradient in the denominator, which is only lower bounded by 0, or some $\epsilon > 0$ at best. Therefore, we directly apply Assumption 4 and $C = \frac{L\eta^2}{2}\|g_t\|_{p^*}^{\frac{2}{p-1}} \leq \frac{L\eta^2 G^{\frac{2}{p-1}}}{2}$. Moving term $A$ to the left hand side, telescoping across iterations, and dividing both sides by $\eta T$ completes the proof. $\square$

## 4.2. $\ell_p$ acceleration

We would note that for smooth convex optimization, (deterministic) gradient descent can be accelerated to achieve a rate of $O(1/T^2)$. However, for stochastic first-order methods, it has been shown that a) in convex settings, SGD cannot improve upon the standard $O(1/\sqrt{T})$ rate when noise parameter $\sigma$ is large enough (Agarwal et al., 2009), and b) in first-order smooth *non-convex* settings, *SGD cannot be accelerated (in theory)* without additional assumptions (in terms of gradient norm minimization), due to known lower bounds (Arjevani et al., 2023). Nevertheless, standard practical implementations of SGD are frequently designed to introduce *some* notion of acceleration with momentum (e.g., (Bernstein et al., 2018; Sutskever et al., 2013)), "pushing" the converging sequence further along the direction of previous gradients.

In contrast, we take the view of acceleration not as a "pushing" (in the Euclidean sense), but rather as a (dynamic) interpolation of two iterate sequences: one acting from a (primal) steepest descent perspective (line 4 Algorithm 1), while the other functions in a dual capacity (line 5 Algorithm 1). An apparent distinction is that momentum, as a separate functionality, can be applied on top of the acceleration scheme in STACEY$_{(p,2)}$, as demonstrated in lines 3 and 7 of Algorithm 1, for both the steepest descent and the (Euclidean) mirror descent.

A Euclidean-based two-sequence interpolation was adopted by Schedule-Free SGD/AdamW (Defazio et al., 2024), which removes explicit learning-rate schedules while retaining strong performance. In the realm of non-Euclidean methods, we contrast our algorithm with Lion-$\mathcal{K}$ (Chen et al., 2024; Bernstein et al., 2018). While at first glance it may seem that these methods may simply be a rewriting of each other (based on the choice of parameters), a closer inspection on *the very first step* reveals that such is not the case:

$$\text{Lion-}\mathcal{K}:\ \theta_1 = -\eta\nabla\mathcal{K}\left((1-\beta_1)\tilde{g}_0\right),$$
$$\text{STACEY}_{(p,2)}:\ \theta_1 = -(1-\tau)\eta s^\epsilon\left((1-\beta_1)\tilde{g}_0\right)$$
$$- \tau\alpha(1-\beta_1)\tilde{g}_0.$$

where $\mathcal{K}(\cdot) = \|\cdot\|_{p^*}$ and $s^\epsilon(\cdot)$ is defined in Algorithm 1. The key difference of STACEY$_{(p,2)}$ lies in the convex combination of a steepest descent step and a gradient descent step, whereas Lion-$\mathcal{K}$ is composed of only the steepest descent step. They coincide only when $\tau = 0$ for STACEY$_{(p,2)}$, i.e., completely getting rid of the "coupling", which then defeats the purpose of our acceleration. In addition, there is no choice of parameters for Lion-$\mathcal{K}$ to recover linear coupling. As a result, they are not iterate-equivalent, which further highlights the fundamental difference between "momentum" and "acceleration", a distinction which, crucially, does not appear in the case of standard (Euclidean) AGD, i.e., when both steepest and mirror descent steps are with respect to Euclidean norms.

Further inspired by the fact that STACEY$_{(p,2)}$ breaks the symmetry (in primal and dual trajectories) by coupling an $\ell_p$ steepest descent step with an $\ell_2$-based mirror descent step, we present the natural variant STACEY$_{(p,p)}$ (Algorithm 2), for which we group $\ell_p$ steepest descent with a mirror descent step having $\frac{1}{p}\|\cdot\|_p^p$ (whose $p^{th}$-order uniform convexity is useful for non-Euclidean acceleration (Adil et al., 2024; Contreras et al., 2024; Song et al., 2021)) as its distance generating function. The closed-form mirror descent update is presented in line 5 of the algorithm.

## 5. Experiments

In this section, we present empirical evidence that the STACEY optimizer outperforms other optimizers in both convergence speed and accuracy. We evaluate STACEY's effectiveness on image classification (Section 5.1), and LLM

*Table 1.* Image classification on CIFAR at the 50th, 100th, and 200th epochs. STACEY consistently outperforms other optimizers, demonstrating both superior accuracy and faster convergence.

| Optimizer | Training NLL ↓ | | | Testing ACC (%) ↑ | | |
|---|---|---|---|---|---|---|
| | @50 epoch | @100 epoch | @200 epoch | @50 epoch | @100 epoch | @200 epoch |
| SGD w/ Momentum | $0.0567 \pm 0.0017$ | $0.0441 \pm 0.0014$ | $0.0352 \pm 0.0012$ | $91.15 \pm 0.30$ | $92.02 \pm 0.24$ | $92.76 \pm 0.13$ |
| Adam | $\mathbf{0.0401} \pm 0.0017$ | $0.0182 \pm 0.0017$ | $0.0083 \pm 0.0010$ | $91.69 \pm 0.18$ | $92.13 \pm 0.16$ | $92.66 \pm 0.36$ |
| AdamW | $0.0590 \pm 0.0010$ | $0.0278 \pm 0.0009$ | $0.0195 \pm 0.0015$ | $90.59 \pm 0.37$ | $91.47 \pm 0.42$ | $92.12 \pm 0.07$ |
| Lion (Chen et al., 2023) | $0.1006 \pm 0.0571$ | $0.2226 \pm 0.1410$ | $0.0245 \pm 0.0043$ | $89.38 \pm 2.02$ | $89.19 \pm 1.88$ | $92.15 \pm 0.32$ |
| STACEY$_{(p,p)}$ | $0.0423 \pm 0.0009$ | $\mathbf{0.0118} \pm 0.0014$ | $0.0021 \pm 0.0011$ | $\mathbf{91.88} \pm 0.21$ | $\mathbf{92.79} \pm 0.16$ | $\mathbf{93.79} \pm 0.38$ |
| STACEY$_{(p,2)}$ | $0.0614 \pm 0.0031$ | $0.0131 \pm 0.0027$ | $\mathbf{0.0014} \pm 0.0005$ | $90.83 \pm 0.32$ | $92.70 \pm 0.28$ | $93.54 \pm 0.06$ |

*Table 2.* Image classification on ImageNet at the 20th, 40th, and 60th epochs. STACEY demonstrates superior test accuracy and faster convergence compared to other optimizers.

| Optimizer | Training NLL ↓ | | | Testing Top-1 ACC (%) ↑ | | |
|---|---|---|---|---|---|---|
| | @20 epoch | @40 epoch | @60 epoch | @20 epoch | @40 epoch | @60 epoch |
| SGD w/ Momentum | $2.0731 \pm 0.0007$ | $1.7926 \pm 0.0006$ | $1.4993 \pm 0.0003$ | $56.34 \pm 0.27$ | $63.54 \pm 0.09$ | $68.81 \pm 0.54$ |
| AdamW | $\mathbf{1.3337} \pm 0.0008$ | $\mathbf{0.9822} \pm 0.0017$ | $\mathbf{0.7395} \pm 0.0029$ | $66.12 \pm 0.53$ | $68.47 \pm 0.14$ | $69.31 \pm 0.05$ |
| Lion (Chen et al., 2023) | $1.3529 \pm 0.0007$ | $1.0948 \pm 0.0126$ | $0.8605 \pm 0.0045$ | $\mathbf{67.66} \pm 0.03$ | $68.43 \pm 0.10$ | $69.62 \pm 0.11$ |
| STACEY$_{(p,p)}$ | $1.4680 \pm 0.0150$ | $1.1636 \pm 0.0159$ | $1.0324 \pm 0.0100$ | $66.93 \pm 0.10$ | $\mathbf{69.15} \pm 0.15$ | $\mathbf{69.87} \pm 0.14$ |
| STACEY$_{(p,2)}$ | $1.8376 \pm 0.0134$ | $1.3781 \pm 0.0187$ | $1.1983 \pm 0.0120$ | $60.89 \pm 0.12$ | $66.34 \pm 0.16$ | $67.56 \pm 0.15$ |

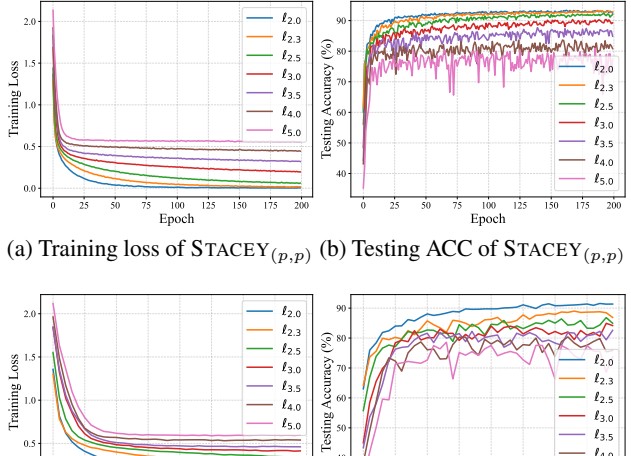

(a) Training loss of STACEY$_{(p,p)}$ (b) Testing ACC of STACEY$_{(p,p)}$

(c) Training loss of STACEY$_{(p,2)}$ (d) Testing ACC of STACEY$_{(p,2)}$

*Figure 1.* Learning curves of CIFAR classification with varying $\ell_p$-norm.

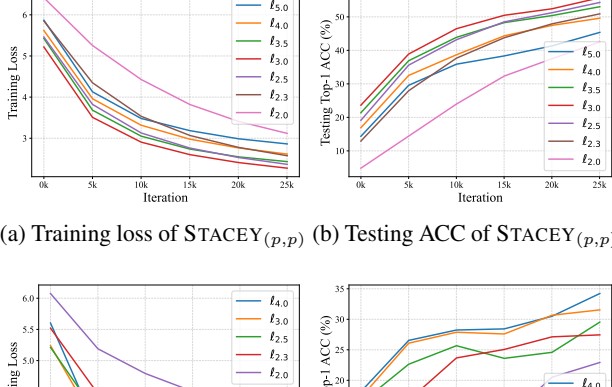

(a) Training loss of STACEY$_{(p,p)}$ (b) Testing ACC of STACEY$_{(p,p)}$

(c) Training loss of STACEY$_{(p,2)}$ (d) Testing ACC of STACEY$_{(p,2)}$

*Figure 2.* Learning curves of ImageNet classification at the first 6 epochs with varying $\ell_p$-norm.

pretraining (Section 5.2). The hyperparameter choices and tuning are summarized in Appendix C.

In all experiments, we underscore the efficiency of the STACEY optimizer by comparing it against other optimizers as baselines including SGD (with momentum) (Nesterov, 1983; Polyak, 1964), Adam (Kingma & Ba, 2015), AdamW (Loshchilov & Hutter, 2019), and Lion (Chen et al., 2023).

In real-world large datasets, such as training from scratch on ImageNet (Deng et al., 2009) and LLM (LLaMA (Touvron et al., 2023)) pretraining on C4 dataset, we further demonstrate the necessity of utilizing different $\ell_p$-norms for specific tasks. For example, in the CIFAR (Krizhevsky, 2009) image classification, an $\ell_p$-norm for $p$ close to 2 delivers the best performance (Section 5.1), consistent with the effectiveness of Euclidean-based optimizers. In contrast, an $\ell_p$-norm with $p$ around 3 proves more effective in LLM

*Table 3.* Training and testing loss of LLM pre-training at a series of steps. The proposed STACEY optimizer consistently achieves lower loss than baselines at all steps.

| Optimizer | Training Loss | | | | Testing Loss | | | |
|---|---|---|---|---|---|---|---|---|
| | @5k step | @10k steps | @20k steps | @30k steps | @5k step | @10k steps | @20k steps | @30k steps |
| SGD w/ Momentum | $6.6704 \pm 0.0129$ | $6.5205 \pm 0.0088$ | $6.4206 \pm 0.0055$ | $6.3920 \pm 0.0048$ | $6.6558 \pm 0.0131$ | $6.5150 \pm 0.0085$ | $6.4173 \pm 0.0038$ | $6.3909 \pm 0.0038$ |
| Adam | $6.4548 \pm 0.0028$ | $6.3647 \pm 0.0037$ | $6.2851 \pm 0.0030$ | $6.2485 \pm 0.0028$ | $6.4493 \pm 0.0017$ | $6.3646 \pm 0.0035$ | $6.2820 \pm 0.0037$ | $6.2480 \pm 0.0028$ |
| AdamW | $5.6655 \pm 0.0095$ | $5.5172 \pm 0.0081$ | $5.4401 \pm 0.0091$ | $5.4268 \pm 0.0096$ | $5.6510 \pm 0.0099$ | $5.5171 \pm 0.0080$ | $5.4350 \pm 0.0088$ | $5.4240 \pm 0.0093$ |
| Lion (Chen et al., 2023) | $6.8722 \pm 0.0656$ | $6.8190 \pm 0.0549$ | $6.8021 \pm 0.0451$ | $6.7794 \pm 0.0425$ | $6.8624 \pm 0.0587$ | $6.8220 \pm 0.0500$ | $6.7954 \pm 0.0438$ | $6.7733 \pm 0.0413$ |
| STACEY$_{(p,p)}$ | $\mathbf{5.4016 \pm 0.0107}$ | $\mathbf{4.9938 \pm 0.0209}$ | $\mathbf{4.6492 \pm 0.0112}$ | $\mathbf{4.4962 \pm 0.0123}$ | $\mathbf{5.3616 \pm 0.0068}$ | $\mathbf{4.9655 \pm 0.0169}$ | $\mathbf{4.6372 \pm 0.0116}$ | $\mathbf{4.4879 \pm 0.0132}$ |
| STACEY$_{(p,2)}$ | $6.2492 \pm 0.0060$ | $6.0038 \pm 0.0319$ | $5.7210 \pm 0.0363$ | $5.5841 \pm 0.0379$ | $6.2312 \pm 0.0065$ | $5.9867 \pm 0.0313$ | $5.7062 \pm 0.0375$ | $5.5755 \pm 0.0375$ |

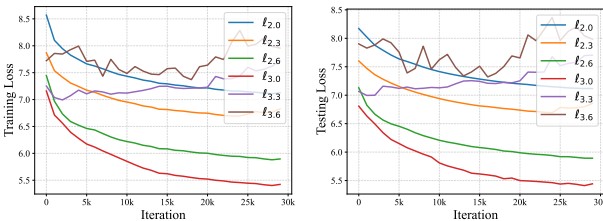

(a) Training loss of STACEY$_{(p,p)}$  (b) Testing loss of STACEY$_{(p,p)}$

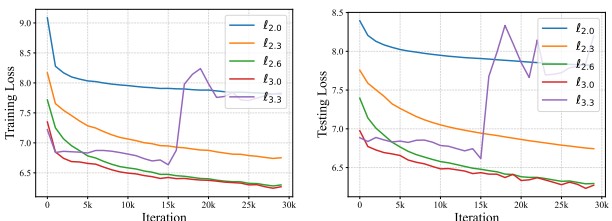

(c) Training loss of STACEY$_{(p,2)}$  (d) Testing loss of STACEY$_{(p,2)}$

*Figure 3.* Learning curves of LLM pretraining at the first 30K iterations with varying $\ell_p$-norm.

pretraining (Section 5.2). These results highlight the importance of developing non-Euclidean optimizers and adjusting the choice of $\ell_p$-norm to enhance performance across different tasks, and we would note this choice may further benefit from, e.g., parameter-free approaches (Jacobsen & Cutkosky, 2022).

## 5.1. Image Classification

We demonstrate improved accuracy and faster convergence of the STACEY optimizer across image classification tasks of varying scales, consistent with our algorithm's design for acceleration.

**Training from scratch on CIFAR.** We train ResNet18 (He et al., 2016) on the CIFAR dataset (Krizhevsky, 2009) for 200 epochs, with the results presented in Table 1. We report training NLL and testing accuracy at the 50th, 100th, and 200th epochs. The proposed STACEY optimizer consistently outperforms all compared optimizers. As shown in Fig. 1, a $p$-norm of 2

yields the best performance for the CIFAR dataset when using the ResNet18 architecture.

**Training from scratch on ImageNet.** We train ResNet50 (He et al., 2016) with a batch size 256 on ImageNet (Deng et al., 2009) for 60 epochs.[3] The learning rate schedule is cosine decay with 10K steps of warm-up, and the mix-precision training is used to reduce the memory footprint. The learning curves are shown in Table 2.

## 5.2. Pretraining Large Language Models (LLMs)

We pretrain llama-100m (Touvron et al., 2023) on the C4 subset[4] using various optimizers with cosine scheduler. The training and testing loss results, as presented in Table 3, show the advantage of STACEY over alternative algorithms. We additionally compare in Fig. 3 the performance of STACEY across different choices of $p$, whereby we observe the best performance when $p = 3$, which contrasts with the best results being observed when $p = 2$ in the CIFAR image classification tasks, as discussed in Section 5.1.

## 5.3. Discussion

As we observe throughout the experiments, STACEY demonstrates superior performance over SGD, which showcases its ability to adapt to a broader range of non-Euclidean geometries. This adaptability verifies STACEY's convergence for general $\ell_p$-norms, making it a better choice for optimization tasks that present complex geometries and extend beyond the conventional Euclidean frameworks.

Compared with Adam (Kingma & Ba, 2015) and AdamW (Loshchilov & Hutter, 2019), the results of STACEY suggests that the introduced acceleration technique is well-aligned with the principles of non-Euclidean optimization. In addition, they highlight how STACEY's acceleration mechanism, which is designed for a wider range of non-Euclidean structure, can yield better performance than tradi-

---

[3]Due to computational resource limitations, the batch sizes used in this paper are smaller than those in Lion's original paper (Chen et al., 2024).

[4]https://huggingface.co/datasets/datablations/c4-subsets.

tional adaptive methods.

Furthermore, STACEY's improved performance over Lion (Chen et al., 2023) highlights the effectiveness of interpolating primal and dual sequences as an acceleration strategy, in contrast to simply incorporating momentum. The primal-dual interpolation ensures a more balanced and stable progression towards optimality, leveraging information from both primal and dual sequences. This strategy allows STACEY to achieve faster convergence, even in challenging settings and complex tasks like large-scale image classification and pretraining LLMs.

**Algorithmic efficiency.** We observe that STACEY has a $2d$ memory overhead, as it needs to store both a momemtum and a dual vector. This matches the memory overhead of Adam, which requires storing two moment vectors, and the per-iteration cost, in terms of basic arithmetic operations, is also comparable to that of Adam. Whereas methods such as SGD with momentum and Lion require only a single momentum vector, we would note that the overhead of the additional dual variable in STACEY is precisely what enables its $\ell_p$-based acceleration.

## 6. Conclusion

In this paper, we have presented a new approach to stochastic non-convex optimization by leveraging *non-Euclidean* $\ell_p$ geometry. We first established that *stochastic $\ell_p$ steepest descent* converges at a rate of $O(\epsilon^{-4})$ in expectation to a stationary point under $\ell_p$-smoothness assumptions, thus strictly generalizing previous analyses for signSGD ($p = \infty$). Building on these foundations, we introduced STACEY, an *accelerated* algorithm that combines stochastic $\ell_p$ descent with primal-dual interpolation techniques to effectively navigate non-Euclidean optimization landscapes.

Our results highlight how acceleration in $\ell_p$ spaces can yield improved geometry-dependent performance compared to Euclidean and $\ell_\infty$-based updates. In extensive experiments on large-scale image classification and language modeling, STACEY consistently achieved faster convergence and higher accuracy than popular optimizers such as SGD, AdamW, and Lion. Moreover, we demonstrated the versatility of choosing different $p \in (2, \infty)$ to tailor the descent geometry to diverse model architectures and datasets. Overall, our contributions underscore both the theoretical and practical benefits of pursuing *non-Euclidean* perspectives for addressing the complexities of modern machine learning tasks.

## Acknowledgements

We thank Jincheng Zhou for helpful discussions related to the experiments implementation. Petros Drineas was partially supported by NSF AF 2209509 and NSF CDSE 2152687.

## Impact Statement

This paper presents work whose goal is to advance the field of Machine Learning. There are many potential societal consequences of our work, none of which we feel must be specifically highlighted here.

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

# A. Proofs

## A.1. Complete Proof for Theorem 1

**Theorem 1** *Running Algorithm 3 on some (possibly non-convex) function $f$ that satisfies Assumptions 1 to 4 yields*

$$\mathbb{E}\left[\frac{1}{T}\sum_{t=0}^{T-1}\|g_t\|_{p^*}^{p^*}\right] \le \frac{f_0 - f^*}{\eta T} + \frac{L\eta G^{\frac{2}{p-1}}}{2} + \frac{1}{T}\sum_{t=0}^{T-1}\frac{\frac{2p-1}{p-1}G^{\frac{1}{p-1}}\|\vec{\sigma}\|_1}{\sqrt{n_t}}$$

*where $f_0 = f(\theta_0)$ and $f^* = f(\theta^*)$, $n_t$ is the batch size in iteration $t$ and $L$, $\vec{\sigma}$, and $G$ are constants from Assumption 1, 3, 4. Further letting the batch size $n_t = T$, the number of gradient call is $N = T^2$ for $T$ iterations. With $\eta = \frac{1}{L^{\frac{1}{2}}G^{\frac{1}{p-1}}T^{\frac{1}{2}}}$ we have*

$$\mathbb{E}\left[\frac{1}{T}\sum_{t=0}^{T-1}\|g_t\|_{p^*}^{p^*}\right] \le \frac{1}{N^{\frac{1}{4}}}\left[L^{\frac{1}{2}}G^{\frac{1}{p-1}}\left(f_0 - f^* + \frac{1}{2}\right) + \frac{2p-1}{p-1}G^{\frac{1}{p-1}}\|\vec{\sigma}\|_1\right],$$

*i.e., Algorithm 3 takes $N \in \mathcal{O}\left(\epsilon^{-4}\right)$ gradient queries to reach an $\epsilon$-approximate stationary point.*

*Proof.* Starting with Assumption 1 and the descent step in Algorithm 3,

$$f(\theta_{t+1}) \le f(\theta_t) + \langle g_t, \theta_{t+1} - \theta_t \rangle + \frac{L}{2}\|\theta_{t+1} - \theta_t\|_p^2$$

$$= f(\theta_t) + \eta \langle g_t, -s(\tilde{g}_t) \rangle + \frac{L}{2}\|s(\tilde{g}_t)\|_p^2$$

$$= f(\theta_t) - \underbrace{\eta \langle g_t, s(g_t) \rangle}_{A} + \underbrace{\eta \langle g_t, s(g_t) - s(\tilde{g}_t) \rangle}_{B} + \underbrace{\frac{L\eta^2}{2}\|s(\tilde{g}_t)\|_p^2}_{C}$$

Now we analyze these terms one by one.

$$A = \sum_{i=1}^{d} g_t^{(i)} \cdot \frac{g_t^{(i)}}{|g_t^{(i)}|^{\frac{p-2}{p-1}}}$$

$$= \sum_{i=1}^{d} |g_t^{(i)}|^{\frac{p}{p-1}}$$

$$= \|g_t\|_{p^*}^{p^*}$$

For term $B$,

$$B = \eta \sum_{i=1}^{d} g_t^{(i)} \left(\frac{g_t^{(i)}}{|g_t^{(i)}|^{\frac{p-2}{p-1}}} - \frac{\tilde{g}_t^{(i)}}{|\tilde{g}_t^{(i)}|^{\frac{p-2}{p-1}}}\right)$$

$$= \eta \sum_{i=1}^{d} g_t^{(i)} \left(\text{sgn}\left(g_t^{(i)}\right) |g_t^{(i)}|^{\frac{1}{p-1}} - \text{sgn}\left(\tilde{g}_t^{(i)}\right) |\tilde{g}_t^{(i)}|^{\frac{1}{p-1}}\right)$$

$$\le \eta \sum_{i=1}^{d} \left|g_t^{(i)}\right| \left|\text{sgn}\left(g_t^{(i)}\right) |g_t^{(i)}|^{\frac{1}{p-1}} - \text{sgn}\left(\tilde{g}_t^{(i)}\right) |\tilde{g}_t^{(i)}|^{\frac{1}{p-1}}\right|$$

$$= \underbrace{\eta \sum_{i=1}^{d} \left|g_t^{(i)}\right| \left(|g_t^{(i)}|^{\frac{1}{p-1}} + |\tilde{g}_t^{(i)}|^{\frac{1}{p-1}}\right) \mathbb{I}_{\left[\text{sgn}\left(g_t^{(i)}\right) \ne \text{sgn}\left(\tilde{g}_t^{(i)}\right)\right]}}_{B_1}$$

$$+ \underbrace{\eta \sum_{i=1}^{d} \left|g_t^{(i)}\right| \left||g_t^{(i)}|^{\frac{1}{p-1}} - |\tilde{g}_t^{(i)}|^{\frac{1}{p-1}}\right| \mathbb{I}_{\left[\text{sgn}\left(g_t^{(i)}\right) = \text{sgn}\left(\tilde{g}_t^{(i)}\right)\right]}}_{B_2}$$

$B_1$ is bounded in expectation by $\frac{2\eta G^{\frac{1}{p-1}}\|\vec{\sigma}\|_1}{\sqrt{n_t}}$ in Lemma 2 and $B_2$ is bounded in expectation by $\frac{\eta G^{\frac{1}{p-1}}\|\vec{\sigma}\|_1}{(p-1)\sqrt{n_t}}$ in Lemma 3.

$$
\begin{aligned}
C &= \frac{L\eta^2}{2}\left(\sum_{i=1}^{d}\left|\frac{g_t^{(i)}}{|g_t^{(i)}|^{\frac{p-2}{p-1}}}\right|^p\right)^{\frac{2}{p}} \\
&= \frac{L\eta^2}{2}\left(\sum_{i=1}^{d}\left|g_t^{(i)}\right|^{\frac{p}{p-1}}\right)^{\frac{2}{p}} \\
&= \frac{L\eta^2}{2}\|g_t\|_{p^*}^{\frac{2}{p-1}} \\
&\leq \frac{L\eta^2 G^{\frac{2}{p-1}}}{2}
\end{aligned}
$$

Therefore,

$$
\eta\mathbb{E}\left[\|g_t\|_{p^*}^{p^*}\right] \leq f(\theta_t) - f(\theta_{t+1}) + \frac{\eta(2p-1)G^{\frac{1}{p-1}}\|\vec{\sigma}\|_1}{(p-1)\sqrt{n_t}} + \frac{L\eta^2 G^{\frac{2}{p-1}}}{2}
$$

By telescoping through $t = 0, \cdots, T-1$, we get

$$
\mathbb{E}\left[\frac{1}{T}\sum_{t=0}^{T-1}\|g_t\|_{p^*}^{p^*}\right] \leq \frac{f(\theta_0) - f(\theta_T)}{\eta T} + \frac{1}{T}\sum_{t=0}^{T-1}\frac{(2p-1)G^{\frac{1}{p-1}}\|\vec{\sigma}\|_1}{(p-1)\sqrt{n_t}} + \frac{L\eta G^{\frac{2}{p-1}}}{2}
$$

$\square$

**Lemma 2.**

$$
\mathbb{E}\left[\eta\sum_{i=1}^{d}\left|g_t^{(i)}\right|\left(|g_t^{(i)}|^{\frac{1}{p-1}} + |\tilde{g}_t^{(i)}|^{\frac{1}{p-1}}\right)\mathbb{I}_{\left[\text{sgn}\left(g_t^{(i)}\right)\neq\text{sgn}\left(\tilde{g}_t^{(i)}\right)\right]}\right] \leq \frac{2\eta G^{\frac{1}{p-1}}\|\vec{\sigma}\|_1}{\sqrt{n_t}}
$$

*Proof.* By Corollary 3 (b),

$$
\begin{aligned}
&\mathbb{E}\left[\eta\sum_{i=1}^{d}\left|g_t^{(i)}\right|\left(|g_t^{(i)}|^{\frac{1}{p-1}} + |\tilde{g}_t^{(i)}|^{\frac{1}{p-1}}\right)\mathbb{I}_{\left[\text{sgn}\left(g_t^{(i)}\right)\neq\text{sgn}\left(\tilde{g}_t^{(i)}\right)\right]}\right] \\
&\leq 2\eta G^{\frac{1}{p-1}}\mathbb{E}\left[\sum_{i=1}^{d}\left|g_t^{(i)}\right|\mathbb{I}_{\left[\text{sgn}\left(g_t^{(i)}\right)\neq\text{sgn}\left(\tilde{g}_t^{(i)}\right)\right]}\right] \\
&= 2\eta G^{\frac{1}{p-1}}\sum_{i=1}^{d}\left|g_t^{(i)}\right|\mathbb{P}\left[\text{sgn}\left(g_t^{(i)}\right)\neq\text{sgn}\left(\tilde{g}_t^{(i)}\right)\right] \\
&\leq 2\eta G^{\frac{1}{p-1}}\sum_{i=1}^{d}\left|g_t^{(i)}\right|\mathbb{P}\left[\left|\tilde{g}_t^{(i)} - g_t^{(i)}\right| \geq \left|g_t^{(i)}\right|\right] \\
&\leq 2\eta G^{\frac{1}{p-1}}\sum_{i=1}^{d}\left|g_t^{(i)}\right|\frac{\mathbb{E}\left[\left|\tilde{g}_t^{(i)} - g_t^{(i)}\right|\right]}{\left|g_t^{(i)}\right|} \\
&\leq 2\eta G^{\frac{1}{p-1}}\sum_{i=1}^{d}\sqrt{\mathbb{E}\left[\left|\tilde{g}_t^{(i)} - g_t^{(i)}\right|^2\right]} \\
&\leq \frac{2\eta G^{\frac{1}{p-1}}\sum_{i=1}^{d}\sigma_i}{\sqrt{n_t}} \\
&= \frac{2\eta G^{\frac{1}{p-1}}\|\vec{\sigma}\|_1}{\sqrt{n_t}}
\end{aligned}
$$

where for the last three inequalities we used Markov's inequality, Jensen's inequality, and Assumption 3. $\square$

**Lemma 3.** $\mathbb{E}\left[\eta \sum_{i=1}^{d}\left|g_t^{(i)}\right|\left|\left|g_t^{(i)}\right|^{\frac{1}{p-1}} - \left|\tilde{g}_t^{(i)}\right|^{\frac{1}{p-1}}\right|\mathbb{I}_{\left[\mathrm{sgn}\left(g_t^{(i)}\right)=\mathrm{sgn}\left(\tilde{g}_t^{(i)}\right)\right]}\right] \le \frac{\eta G^{\frac{1}{p-1}}\|\vec{\sigma}\|_1}{(p-1)\sqrt{n_t}}.$

*Proof.* Denoting $\mathbb{E}\left[\cdot \mid \mathrm{sgn}\left(g_t^{(i)}\right) = \mathrm{sgn}\left(\tilde{g}_t^{(i)}\right)\right]$ as $\mathbb{E}_{|=}[\cdot]$, and $\mathbb{P}\left[\mathrm{sgn}\left(g_t^{(i)}\right) = \mathrm{sgn}\left(\tilde{g}_t^{(i)}\right)\right]$ as $\mathbb{P}[=]$,

$$\mathbb{E}\left[\eta \sum_{i=1}^{d}\left|g_t^{(i)}\right|\left|\left|g_t^{(i)}\right|^{\frac{1}{p-1}} - \left|\tilde{g}_t^{(i)}\right|^{\frac{1}{p-1}}\right|\mathbb{I}_{\left[\mathrm{sgn}\left(g_t^{(i)}\right)=\mathrm{sgn}\left(\tilde{g}_t^{(i)}\right)\right]}\right]$$

$$= \eta \mathbb{E}_{|=}\left[\sum_{i=1}^{d}\left|g_t^{(i)}\right|\left|\left|g_t^{(i)}\right|^{\frac{1}{p-1}} - \left|\tilde{g}_t^{(i)}\right|^{\frac{1}{p-1}}\right|\right]\mathbb{P}[=]$$

$$= \eta \mathbb{E}_{|=}\left[\sum_{i=1}^{d}\left|g_t^{(i)}\right|\left|\left|g_t^{(i)}\right|^{\frac{1}{p-1}} - \left|\tilde{g}_t^{(i)}\right|^{\frac{1}{p-1}}\right|\right]\mathbb{P}[=]$$

$$= \eta \mathbb{E}_{|=}\left[\sum_{i=1}^{d}\left|g_t^{(i)}\right|\left|\left(\left|\tilde{g}_t^{(i)}\right|^{\frac{1}{p-1}} + \frac{1}{p-1}\mathrm{sgn}(\zeta^{(i)})\left|\zeta^{(i)}\right|^{\frac{2-p}{p-1}}\left(g_t^{(i)} - \tilde{g}_t^{(i)}\right)\right) - \left|\tilde{g}_t^{(i)}\right|^{\frac{1}{p-1}}\right|\right]\mathbb{P}[=]$$

$$= \eta \mathbb{E}_{|=}\left[\sum_{i=1}^{d}\left|g_t^{(i)}\right|\left|\frac{1}{p-1}\mathrm{sgn}(\zeta^{(i)})\left|\zeta^{(i)}\right|^{\frac{2-p}{p-1}}\left(g_t^{(i)} - \tilde{g}_t^{(i)}\right)\right|\right]\mathbb{P}[=]$$

$$= \frac{\eta}{p-1}\mathbb{E}_{|=}\left[\sum_{i=1}^{d}\left|g_t^{(i)}\right|\left|\zeta^{(i)}\right|^{\frac{2-p}{p-1}}\left|g_t^{(i)} - \tilde{g}_t^{(i)}\right|\right]\mathbb{P}[=],$$

in which the second equality holds by taking the zeroth order Taylor expansion of $\left|g_t^{(i)}\right|^{\frac{1}{p-1}}$ at $\tilde{g}_t^{(i)}$ with Lagrange remainder, and $\zeta^{(i)}$ is in the range from $g_t^{(i)}$ to $\tilde{g}_t^{(i)}$. Given $\mathrm{sgn}\left(g_t^{(i)}\right) = \mathrm{sgn}\left(\tilde{g}_t^{(i)}\right)$, by the definition of $\zeta^{(i)}$ in the Lagrange remainder, we must have either $\left|g_t^{(i)}\right| \le \left|\zeta^{(i)}\right| \le \left|\tilde{g}_t^{(i)}\right|$ or $\left|g_t^{(i)}\right| \ge \left|\zeta^{(i)}\right| \ge \left|\tilde{g}_t^{(i)}\right|$. Now we analyze these two cases respectively. We write out the derivations separately for clarity and simplicity, alternatively one can merge these two cases with the law of total expectation.

(1) If $\left|g_t^{(i)}\right| \le \left|\zeta^{(i)}\right| \le \left|\tilde{g}_t^{(i)}\right|$, then

$$\frac{\eta}{p-1}\mathbb{E}_{|=}\left[\sum_{i=1}^{d}\left|g_t^{(i)}\right|\left|\zeta^{(i)}\right|^{\frac{2-p}{p-1}}\left|g_t^{(i)} - \tilde{g}_t^{(i)}\right|\right]\mathbb{P}[=] \le \frac{\eta}{p-1}\mathbb{E}_{|=}\left[\sum_{i=1}^{d}\left|\zeta^{(i)}\right|\left|\zeta^{(i)}\right|^{\frac{2-p}{p-1}}\left|g_t^{(i)} - \tilde{g}_t^{(i)}\right|\right]\mathbb{P}[=]$$

$$= \frac{\eta}{p-1}\mathbb{E}_{|=}\left[\sum_{i=1}^{d}\left|\zeta^{(i)}\right|^{\frac{1}{p-1}}\left|g_t^{(i)} - \tilde{g}_t^{(i)}\right|\right]\mathbb{P}[=]$$

$$\le \frac{\eta}{p-1}\mathbb{E}_{|=}\left[\sum_{i=1}^{d}\left|\tilde{g}_t^{(i)}\right|^{\frac{1}{p-1}}\left|g_t^{(i)} - \tilde{g}_t^{(i)}\right|\right]\mathbb{P}[=]$$

$$\le \frac{\eta G^{\frac{1}{p-1}}}{p-1}\sum_{i=1}^{d}\mathbb{E}_{|=}\left[\left|g_t^{(i)} - \tilde{g}_t^{(i)}\right|\right]\mathbb{P}[=]$$

$$= \frac{\eta G^{\frac{1}{p-1}}}{p-1}\sum_{i=1}^{d}\frac{\mathbb{E}\left[\left|g_t^{(i)} - \tilde{g}_t^{(i)}\right|\right]}{\mathbb{P}[=]}\mathbb{P}[=]$$

$$\le \frac{\eta G^{\frac{1}{p-1}}}{p-1}\sum_{i=1}^{d}\sqrt{\mathbb{E}\left[\left|g_t^{(i)} - \tilde{g}_t^{(i)}\right|^2\right]} \qquad \text{(Jensen's)}$$

$$\le \frac{\eta G^{\frac{1}{p-1}}}{p-1}\sum_{i=1}^{d}\frac{\sigma_i}{\sqrt{n_t}} \qquad \text{(Assumption 3)}$$

$$= \frac{\eta G^{\frac{1}{p-1}}\|\vec{\sigma}\|_1}{(p-1)\sqrt{n_t}}$$

(2) If $\left|g_t^{(i)}\right| \geq \left|\zeta^{(i)}\right| \geq \left|\tilde{g}_t^{(i)}\right|$, then

$$\frac{\eta}{p-1}\mathbb{E}_{|=}\left[\sum_{i=1}^{d}\left|g_t^{(i)}\right|\left|\zeta^{(i)}\right|^{\frac{2-p}{p-1}}\left|g_t^{(i)}-\tilde{g}_t^{(i)}\right|\right]\mathbb{P}\left[=\right]$$

$$\leq \frac{\eta}{p-1}\mathbb{E}_{|=}\left[\sum_{i=1}^{d}\left|g_t^{(i)}\right|\left|\tilde{g}_t^{(i)}\right|^{\frac{2-p}{p-1}}\left|g_t^{(i)}-\tilde{g}_t^{(i)}\right|\right]\mathbb{P}\left[=\right]$$

$$\leq \frac{\eta}{(p-1)\mathbb{P}[=]}\mathbb{E}\left[\sum_{i=1}^{d}\left|g_t^{(i)}\right|\left|\tilde{g}_t^{(i)}\right|^{\frac{2-p}{p-1}}\left|g_t^{(i)}-\tilde{g}_t^{(i)}\right|\right]\mathbb{P}\left[=\right]$$

$$\leq \frac{\eta}{p-1}\sum_{i=1}^{d}\sqrt{\mathbb{E}\left[\left|g_t^{(i)}\right|^2\left|\tilde{g}_t^{(i)}\right|^{\frac{2(2-p)}{p-1}}\right]\mathbb{E}\left[\left|g_t^{(i)}-\tilde{g}_t^{(i)}\right|^2\right]} \qquad \text{(Cauchy-Schwarz)}$$

$$\leq \frac{\eta}{p-1}\sum_{i=1}^{d}\sqrt{\left|g_t^{(i)}\right|^2\mathbb{E}\left[\left|\tilde{g}_t^{(i)}\right|^{\frac{2(2-p)}{p-1}}\right]\frac{\sigma_i^2}{n_t}} \qquad \text{(Assumption 3)}$$

$$\leq \frac{\eta}{p-1}\sum_{i=1}^{d}\sqrt{\left|g_t^{(i)}\right|^2\left(\mathbb{E}\left[\left|\tilde{g}_t^{(i)}\right|^2\right]\right)^{\frac{2-p}{p-1}}\frac{\sigma_i^2}{n_t}} \qquad \text{(Jensen's)}$$

$$\leq \frac{\eta}{p-1}\sum_{i=1}^{d}\sqrt{\left|g_t^{(i)}\right|^2\left(\mathrm{Var}\left[\tilde{g}_t^{(i)}\right]+\left(\mathbb{E}\left[\tilde{g}_t^{(i)}\right]\right)^2\right)^{\frac{2-p}{p-1}}\frac{\sigma_i^2}{n_t}} \qquad \text{(Variance Definition)}$$

$$\leq \frac{\eta}{p-1}\sum_{i=1}^{d}\sqrt{\left|g_t^{(i)}\right|^2\left(\mathbb{E}\left[\tilde{g}_t^{(i)}\right]\right)^{\frac{2(2-p)}{p-1}}\frac{\sigma_i^2}{n_t}}$$

$$= \frac{\eta}{p-1}\sum_{i=1}^{d}\left|g_t^{(i)}\right|^{\frac{1}{p-1}}\frac{\sigma_i}{\sqrt{n_t}} \qquad \text{(Assumption 2)}$$

$$\leq \frac{\eta G^{\frac{1}{p-1}}\|\vec{\sigma}\|_1}{(p-1)\sqrt{n_t}}.$$

Combining these two cases together (e.g., by the law of total expectation) completes the proof. $\qquad\square$

## B. $\ell_2$ Majorization and $\ell_p$ Smoothness

An assumption of interest, studied by Bernstein et al. (2018) (as well as Karimi et al. (2016)), is that of $\ell_2$ *majorization* (with respect to $\vec{L} = [L_1, \ldots, L_d]$), meaning that for all $x, y \in \mathbb{R}^d$,

$$\left|f(y) - f(x) - \nabla f(x)^\top (y-x)\right| \leq \frac{1}{2}\sum_{i=1}^{d}L_i(y^{(i)} - x^{(i)})^2.$$

We may equivalently express this condition as 1-smoothness w.r.t. $\|\cdot\|_{\mathbf{L}}$, where $\mathbf{L} := \mathrm{diag}(\vec{L})$, i.e., for all $x, y \in \mathbb{R}^d$, $\|\nabla f(y) - \nabla f(x)\|_{\mathbf{L}^{-1}} \leq \|y - x\|_{\mathbf{L}}$.

Interestingly, we may observe that, for any $1 < \rho \leq \infty$ and letting $\rho^* := \frac{\rho}{\rho-1}$, we have

$$\frac{1}{\|\vec{L}\|_{\rho^*}^{1/2}}\|\nabla f(y) - \nabla f(x)\|_{2\rho/(2\rho-1)} \leq \|\nabla f(y) - \nabla f(x)\|_{\mathbf{L}^{-1}} \leq \|y - x\|_{\mathbf{L}} \leq \|\vec{L}\|_{\rho^*}^{1/2}\|y - x\|_{2\rho},$$

where the first inequality holds by reverse Hölder's inequality, i.e., for $u, v \in \mathbb{R}^d$, $\sum_{i=1}^{d}|u^{(i)}v^{(i)}| \geq \|u\|_{1/q}\|v\|_{\frac{-1}{q-1}}$ (where we choose $q = \frac{2\rho-1}{\rho}$), and the last inequality holds by Hölder's inequality.

Rearranging, we have $\|\nabla f(y) - \nabla f(x)\|_{2\rho/(2\rho-1)} \leq \|\vec{L}\|_{\rho^*}\|y - x\|_{2\rho}$, and so it follows that, for $p > 2$, $\ell_2$ majorization implies $\|\vec{L}\|_{\frac{p}{p-2}}$-smoothness w.r.t. $\|\cdot\|_p$. Thus, while this condition is sufficient to entail $\ell_p$ smoothness (as previously noted by (Balles et al., 2020) in the case of $p = \infty$), we nevertheless prefer to work directly with $\ell_p$ smoothness assumptions, as we believe they provide a more natural pairing for the methods we consider.

## C. Hyperparameter Choices

We summarize the hyperparameters used in our experiments in Tables 4, 5, and 6. These hyperparameters are determined through a grid search. Specifically, we perform a search to identify appropriate values for the $\ell_p$-norm, learning rate $\eta$, $\alpha$, and weight decay $\lambda$. This process involves an initial rough comparison across a range of magnitudes, followed by a more precise grid search to determine the optimal values.

For fair comparison, all experimental settings, apart from the listed hyperparameters, follow the original papers of AdamW (Loshchilov & Hutter, 2019) and Lion (Chen et al., 2023), and are kept consistent across all optimizers. For example, data augmentations for ImageNet (Deng et al., 2009) and CIFAR (Krizhevsky, 2009) all include random cropping and random horizontal flipping.

*Table 4.* CIFAR hyper-parameters.

| Model | Optimizer | Batch Size | $p$ | Learning Rate | Schedule | $\alpha$ | $\beta_1$ | $\beta_2$ | $\lambda$ | $\tau$ | $\epsilon$ |
|---|---|---|---|---|---|---|---|---|---|---|---|
| ResNet-18 | SGD w/ Momentum | 128 | - | 0.02 | cosine decay | - | 0.9 | - | 0.0002 | - | - |
| ResNet-18 | Adam (Kingma & Ba, 2015) | 128 | - | 0.001 | cosine decay | - | 0.9 | 0.999 | 0.0005 | - | 1e-8 |
| ResNet-18 | AdamW (Loshchilov & Hutter, 2019) | 128 | - | 0.01 | cosine decay | - | 0.9 | 0.999 | 0.0005 | - | 1e-8 |
| ResNet-18 | Lion (Chen et al., 2023) | 128 | - | 0.001 | cosine decay | - | 0.9 | 0.99 | 0.01 | - | - |
| ResNet-18 | STACEY$_{(p,p)}$ | 128 | 2 | 0.1 | cosine decay | 0.1 | 0.9 | 0.99 | 0.01 | 0.001 | 1e-12 |
| ResNet-18 | STACEY$_{(p,2)}$ | 128 | 2 | 0.1 | cosine decay | 0.1 | 0.9 | 0.99 | 0.01 | 0.001 | 1e-12 |

*Table 5.* ImageNet hyper-parameters.

| Model | Optimizer | Batch Size | $p$ | Learning Rate | Schedule | $\alpha$ | $\beta_1$ | $\beta_2$ | $\lambda$ | $\tau$ | $\epsilon$ |
|---|---|---|---|---|---|---|---|---|---|---|---|
| ResNet-50 | SGD w/ Momentum | 256 | - | 0.01 | cosine decay | - | 0.9 | - | 0.0005 | - | - |
| ResNet-50 | AdamW (Loshchilov & Hutter, 2019) | 256 | - | 0.002 | cosine decay | - | 0.9 | 0.999 | 0.005 | - | 1e-4 |
| ResNet-50 | Lion (Chen et al., 2023) | 256 | - | 3e-4 | cosine decay | - | 0.9 | 0.99 | 0.01 | - | - |
| ResNet-50 | STACEY$_{(p,p)}$ | 256 | 3 | 0.01 | cosine decay | 0.001 | 0.9 | 0.999 | 0.001 | 0.001 | 1e-8 |
| ResNet-50 | STACEY$_{(p,2)}$ | 256 | 2.8 | 0.01 | cosine decay | 0.001 | 0.9 | 0.999 | 0.001 | 0.001 | 1e-8 |

*Table 6.* Hyper-parameters for LLM pretraining.

| Model | Optimizer | Batch Size | $p$ | Learning Rate | Schedule | $\alpha$ | $\beta_1$ | $\beta_2$ | $\lambda$ | $\tau$ | $\epsilon$ |
|---|---|---|---|---|---|---|---|---|---|---|---|
| llama-100m | SGD w/ Momentum | 16 | - | 0.01 | cosine decay | - | 0.9 | - | 0.0005 | - | - |
| llama-100m | Adam (Kingma & Ba, 2015) | 16 | - | 0.0001 | cosine decay | - | 0.9 | 0.999 | 0.01 | - | 1e-8 |
| llama-100m | AdamW (Loshchilov & Hutter, 2019) | 16 | - | 0.0001 | cosine decay | - | 0.9 | 0.999 | 0.05 | - | 1e-8 |
| llama-100m | Lion (Chen et al., 2023) | 16 | - | 0.05 | cosine decay | - | 0.9 | 0.999 | 0.01 | - | - |
| llama-100m | STACEY$_{(p,p)}$ | 16 | 3 | 0.01 | cosine decay | 0.1 | 0.9 | 0.99 | 0.01 | 0.001 | 1e-8 |
| llama-100m | STACEY$_{(p,2)}$ | 16 | 2.8 | 0.01 | cosine decay | 0.1 | 0.9 | 0.99 | 0.0005 | 0.001 | 1e-8 |

