# OpenReview forum: "Stacey: Promoting Stochastic Steepest Descent via Accelerated $\ell_p$-Smooth Nonconvex Optimization"
_ICML.cc/2025/Conference — ICML 2025 poster_

### Official Review · Reviewer_ddGc · 2025-03-12

**Overall Recommendation:** 1

**Summary:**

The paper uses different mixed Lp norms to run SGD

**Claims And Evidence:**

The main takeaway is that there are different Lp norms boost performance of optimization for different problems. For CNN's for example they find L2 to work best but for LLMs they find another L3 to work better for example. Unfortunately there are no error bars or standard deviation in tables. Also the learning rates for the baseline Adam is very off for LLMs, its set to 1e-4 where it should really be set to 1e-3 or higher. Also the epsilon for Adam is also off. Setting epsilon to 1e-8 basically makes Adam act as SGD + M. On LLMs the epsilon should be set closer to 1e-17.

**Essential References Not Discussed:**

NA

**Experimental Designs Or Analyses:**

please see other sections.

**Methods And Evaluation Criteria:**

I believe the baselines were not set correctly.

**Other Comments Or Suggestions:**

Fix baselines.

**Other Strengths And Weaknesses:**

The paper lacks variance bars or standard deviations as well as weakly tuned baselines.

**Questions For Authors:**

NA

**Relation To Broader Scientific Literature:**

NA

**Theoretical Claims:**

The paper does a good job of contextualizing their method in modern optimization.

The paper gives some extensions to simpler claims from but good overall.

[1] Guillaume Garrigos, Robert M. Gower, Handbook of Convergence Theorems for (Stochastic) Gradient Methods

[2] Ahmed Khaled, Peter Richtárik, Better Theory for SGD in the Nonconvex World

---

> ### Author Rebuttal · Authors · 2025-04-01
>
> We thank the reviewer for their comments and suggestions.
>
> >**Error Bar and Standard Deviation**
> > No error bars or standard deviation in tables
>
> We ran 3 random seeds to obtain the error bar.
>
> CIFAR:
>
> | **Optimizer** | **Train NLL @50** | **Train NLL @100** | **Train NLL @200** | **Test ACC @50** | **Test ACC @100** | **Test ACC @200** |
> |---|:---:|:---:|:---:|:---:|:---:|:---:|
> | SGD w/ Momentum | 0.0567 ± 0.0017 | 0.0441 ± 0.0014 | 0.0352 ± 0.0012 | 91.15 ± 0.30 | 92.02 ± 0.24 | 92.76 ± 0.13 |
> | Adam | 0.0401 ± 0.0017 | 0.0182 ± 0.0017 | 0.0083 ± 0.0010 | 91.69 ± 0.18 | 92.13 ± 0.16 | 92.66 ± 0.36 |
> | AdamW | 0.0590 ± 0.0010 | 0.0278 ± 0.0009 | 0.0195 ± 0.0015 | 90.59 ± 0.37 | 91.47 ± 0.42 | 92.12 ± 0.07 |
> | Lion | 0.1006 ± 0.0571 | 0.2226 ± 0.1410 | 0.0245 ± 0.0043 | 89.38 ± 2.02 | 89.19 ± 1.88 | 92.15 ± 0.32 |
> | Stacey(p,p) | 0.0423 ± 0.0009 | 0.0118 ± 0.0014 | 0.0021 ± 0.0011 | 91.88 ± 0.21 | 92.79 ± 0.16 | 93.79 ± 0.38 |
> | Stacey(p,2) | 0.0614 ± 0.0031 | 0.0131 ± 0.0027 | 0.0014 ± 0.0005 | 90.83 ± 0.32 | 92.70 ± 0.28 | 93.54 ± 0.06 |
>
>
> ImageNet:
>
> | **Optimizer** | **Train NLL @20** | **Train NLL @40** | **Train NLL @60** | **Test Top-1 ACC @20** | **Test Top-1 ACC @40** | **Test Top-1 ACC @60** |
> |---|:---:|:---:|:---:|:---:|:---:|:---:|
> | Stacey(p,p) | 1.4680 ± 0.0150 | 1.1636 ± 0.0159 | 1.0324 ± 0.0100 | 66.93 ± 0.10 | 69.15 ± 0.15 | 69.87 ± 0.14 |
>
>
> LLM:
>
> | **Optimizer** | **Train loss @5K** | **Train loss @10K** | **Train loss @20K** | **Train loss @30K** | **Test loss @5K** | **Test loss @10K** | **Test loss @20K** | **Test loss @30K** |
> |---|:---:|:---:|:---:|:---:|:---:|:---:|:---:|:---:|
> | SGD w/ Momentum | 6.6704 ± 0.0129 | 6.5205 ± 0.0088 | 6.4206 ± 0.0055 | 6.3920 ± 0.0048 | 6.6558 ± 0.0131 | 6.5150 ± 0.0085 | 6.4173 ± 0.0038 | 6.3909 ± 0.0038 |
> | Adam | 6.4548 ± 0.0028 | 6.3647 ± 0.0037 | 6.2851 ± 0.0030 | 6.2485 ± 0.0028 | 6.4493 ± 0.0017 | 6.3646 ± 0.0035 | 6.2820 ± 0.0037 | 6.2480 ± 0.0028 |
> | AdamW | 5.6655 ± 0.0095 | 5.5172 ± 0.0081 | 5.4401 ± 0.0091 | 5.4268 ± 0.0096 | 5.6510 ± 0.0099 | 5.5171 ± 0.0080 | 5.4350 ± 0.0088 | 5.4240 ± 0.0093 |
> | Lion | 6.8722 ± 0.0656 | 6.8190 ± 0.0549 | 6.8021 ± 0.0451 | 6.7794 ± 0.0425 | 6.8624 ± 0.0587 | 6.8220 ± 0.0500 | 6.7954 ± 0.0438 | 6.7733 ± 0.0413 |
> | Stacey(p,p) | 5.4016 ± 0.0107 | 4.9938 ± 0.0209 | 4.6492 ± 0.0112 | 4.4962 ± 0.0123 | 5.3616 ± 0.0068 | 4.9655 ± 0.0169 | 4.6372 ± 0.0116 | 4.4879 ± 0.0132 |
> | Stacey(p,2) | 6.2492 ± 0.0060 | 6.0038 ± 0.0319 | 5.7210 ± 0.0363 | 5.5841 ± 0.0379 | 6.2312 ± 0.0065 | 5.9867 ± 0.0313 | 5.7062 ± 0.0375 | 5.5755 ± 0.0375 |
>
>
>
> >**Baseline Settings**
> > Learning rates for the baseline Adam is very off for LLMs; its set to 1e-4 where it should really be set to 1e-3 or higher. Also, the epsilon for Adam is also off. Setting epsilon to 1e-8 basically makes Adam act as SGD + M. On LLMs the epsilon should be set closer to 1e-17.
>
> We would kindly ask the reviewer to provide additional details regarding which LLMs they are referring to, as we did not observe a notable difference using the suggested parameters for our setting.
> | **Optimizer** | **lr** | **eps** | **Test loss @15K** | **Test loss @30K** |
> |:---:|:---:|:---:|:---:|:---:|
> | Adam | 1e-3 | 1e-17 | 6.4120 | 6.2341 |
> | Adam (Our settings) | 1e-4 | 1e-8 | 6.3102 | 6.2485 |
>
> >**Performance Boost**
> >I downloaded the code and ran it on a few benchmarks. I did not see a boost in modded-nanoGPT, which is a properly tuned baseline. I also did not see a boost vs Adam in some toy problems like xor
>
> We would kindly ask the reviewer to elaborate on the details of their evaluation, as otherwise, we are unable to provide a proper response or explanation.

---

> > ### Comment · Reviewer_ddGc · 2025-04-04
> >
> > Thank you for the experiments. I will certainly take them into consideration.
> >
> > Here are the last two. Feel free to tune as much as you like.
> >
> > Modded nanoGPT is the following benchmark. NanoGPT (124M) in 3 minutes. Can the authors run the 124M and the bigger benchmarks and report the results?
> >
> > https://github.com/KellerJordan/modded-nanogpt
> >
> > xor is the following benchmark. I tested the uploaded code, but I cannot seem to get Stacy to outperform Adam.
> >
> > https://github.com/lixilinx/psgd_torch/blob/master/rnn_xor_problem_general_purpose_preconditioner.py

---

> > > ### Author Response · Authors · 2025-04-08
> > >
> > > The results of the 124M benchmark are as follows:
> > >
> > > | **Optimizer** | **Val loss @0.2B tokens** | **Val loss @0.4B tokens** | **Val loss @0.6B tokens** | **Val loss @0.8B tokens** |
> > > |:---:|:---:|:---:|:---:|:---:|
> > > | AdamW | 4.715 | 4.055 | 3.853 | 3.765 |
> > > | Stacey(p,p) | 4.157 | 3.887 | 3.762 | 3.688 |
> > >
> > > We observe a notable improvement over AdamW, which is consistent with the LLM experiments in our paper. We set nearly all of the hyperparameters the same as listed in the paper for the LLM experiments with Stacey(p,p) (Table 7), except for $\alpha = 0.1$ and $\lambda = 0.001$.
> > >
> > > We further wish to emphasize the overall contributions of our work, namely a primal-dual view of $\ell_p$ steepest descent that we justify both theoretically and empirically.
> > >
> > > Having addressed these concerns and provided additional context for our contributions, we kindly ask the reviewer to reconsider their evaluation.

---

### Official Review · Reviewer_d7Gi · 2025-03-13

**Overall Recommendation:** 4

**Summary:**

This paper introduces Stacey an optimisation algorithm targeted at training deep neural networks (DNNs). Stacey generalises SignSGD and conventional SGD, in a p-norm sense, where SGD uses the 2-norm to measure distance and SignSGD uses the inf-norm. On top of this Stacey include a acceleration scheme to aid the speed of convergence. The paper offers a theoretical result, specifically a convergence rate for an non-accelerated version of Stacey on smooth stochastic problems with bounded gradient and variance.  Additionally there is an empirical evaluation of Stacey on standard DNN benchmarks against popular deep learning algorithms.

**Claims And Evidence:**

The theoretical claims seem well supported.

The empirical claims are supported however I do have some doubts about the fairness of the empirical evaluation when comparing to other methods, however it is difficult to be totally objective here, given the natural differences between optimisers. Inevitably the wider community will need to judge how Stacey performs in practice compared to existing methods.

**Essential References Not Discussed:**

This recent paper is missing, and would be great to see included as a baseline, specially given its lower number of hyperparameters.

Defazio A, Yang X, Khaled A, Mishchenko K, Mehta H, Cutkosky A. The road less scheduled. Advances in Neural Information Processing Systems. 2024 Dec 16;37:9974-10007.

**Experimental Designs Or Analyses:**

The experimental design looks okay, though there are definitely some flaws here, see weakness section.

**Methods And Evaluation Criteria:**

The benchmarks look appropriate, it would have been nice to see some smaller networks and more classical optimisation problems considered, given these don't take much compute.

**Other Comments Or Suggestions:**

It would be great to detail the grids searched over in terms of hyperparameters, not just the final values.

Some idea of the robustness of stacey to its hyperparameters would really help sell its practicality.

Please explain why the number of iteration at which results (both tables and plots) are shown seem to vary so much.

**Other Strengths And Weaknesses:**

*Strengths*
1) The paper is well written.
2) The idea behind the paper seems neat and some related theoretical results are provided.
3) The benchmarks considered look appropriate, of course it would be nice to see more, including some more classical non-deep learning optimisation problems.
4) This seems a promising direction for further research to build on top of.

*Weaknesses*
1) The empirical experimental section has some flaws:
i) Some recent baselines are missing from the experimental section, specifically the one introduced last year in "The Road Less Scheduled"
ii) The amount of hyperparameters tuned for Stacey seems to far exceed the other methods, making the comparison hard to judge.
iii) It is not clear how robust Stacey is to its hyperparameters, nor how much hyperparameter tuning is required to get good results.
iv) There is lack of error bars or mention of variation between runs
v) Smaller scale classical optimisation problems are missing
vi) For the ImageNet results It is not clear why the results are presented at epoch 60 rather than the typical epoch 90 given at least some of the experiment were run for 90 epochs, as report in the appendix
vii) Tables of some results seem to be missing final, such as PPL on transformer pretraining.
viii) The number of iterations shown vary between plots for no explained reason.
ix) Some of the results for Stacey(2,2) seem to be different between plots (figure 5&6) & (figure 3 vs 11&12)
x) The transformer pertaining experiments seem to be missing some important details, model architecture what is LLama-100?, why test ppl increases, unexplained differences in performance between plots see point ix.
xi) Missing results for Stochastic ℓp Descent.

2) The totality of the above critics makes me wonder how much the results are being presented in a way to make them seem best. I would suggest spending a little more time in the appendix making it clear why specific choices were made. Without the empirical results the paper doesn't offer enough of a contribution in my opinion so it is essential a reader is not wondering why some seemly odd choice have been made in the way the experiments have been conducted and presented. My score is assuming greater clarity is given on the experiments in the final version of the paper.

3) The theoretical results are not presented for Stacey but a far simpler non-accelerated algorithm.

**Questions For Authors:**

1) Stacey has *a lot* of hyperparameters, many more than typically tuned for SGD and Adam, do you think your comparison is fair given this fact? Did you run a comparison where the same number of limited hyperparameters (say 1 or 2 only) are adjusted for all algorithms and the rest are left at their default values? Much of Adam (&AdamW) success is due to their robustness to it's hyperparamters, algorithms that need extensive hyperparameter tuning are unlikely to be used in practice. The experiments present in the paper do not make it clear to me how useful Stacey is as a practical algorithm.

2) The p norm used in Stacey is a hyperparameter, do you think it would be possible to adept Stacey to automatically work out the best setting of "p" for a given problem during the optimisation process? This would would help reduce the amount of hyperparamters required.

3) Further to the above, I understand the p norm used to measure distance in Stacey is fixed for all parameters, do you think it would be possible to extend stacey so it is adaptive adjusting the p-norm used, say per layer?

4) For some tasks Stacey(p,2) does better and some Stacey(p,p) does better what do you think might be causing this discrepancy?

5) Do you think Stacey with acceleration enjoys any theoretically properties? Did you make any progress to this end?

6) Is Stacey(p,p) equivalent to Stacey(p,2) and Stacey(2,p), If p = 2. If so, why do they seem to behave differently in your experiments?

**Relation To Broader Scientific Literature:**

The Relation To Broader Scientific Literature is well motivated.

**Theoretical Claims:**

I did not thoroughly check the proofs due to time. The Theoretical Claims presented are not for Stacey but a far simpler non-accelerated algorithm.

---

> ### Author Rebuttal · Authors · 2025-04-01
>
> We thank the reviewer for their thoughtful comments and suggestions, and we respond to their questions below.
>
> > **Missing Reference**
>
> Thanks for the suggestion. We will provide a citation and include it as a baseline in the updated manuscript.
>
> > **Hyperparameters**
>
> Though we may tune $\tau$ and $\alpha$, we found that, similar to SGD and Adam, the best choice comes from a small set, i.e., $\tau \in [0.001]$ and $\alpha \in [0.1, 0.01, 0.001]$, and so we believe this provides a fair comparison with other methods. We did run comparisons where a limited number of hyperparameters were tuned, while the rest were left at default values, as such defaults let us reduce the scope of the search.
>
> > **Questions and Comments on Experiments**
> > 1. Lack of error bars/variation
>
> For space reasons, we kindly point the reviewer to our rebuttal for Reviewer ddGc for the tables of results with error bars.
>
> > 2. Smaller scale classical optimisation problems are missing
>
> While our method was designed with large-scale models in mind, we will include smaller scale experiments, and we would kindly ask the reviewer for suggested problems.
>
> > 3. ImageNet result epochs
>
> Thanks for catching this. This is a typo from an outdated table in the appendix, which we will update accordingly.
>
> > 4. Tables of some results seem to be missing final
>
> If we understand correctly, we will include tables of the PPL/transformer pretraining results, with error bars/variance included (as in the tables provided in the rebuttal).
>
> > 5. The number of iterations shown vary
>
> The iteration counts in Figures 2 and 4 differ because they represent two distinct experimental setups: one for ImageNet classification, the other for LLM pretraining, each with its own training configuration and iteration schedule.
>
> > 6.  x) The transformer pertaining experiments details
>
> LLama-100 is adopted from the github repo of "GaLore: Memory-Efficient LLM Training by Gradient Low-Rank Projection" [Zhao et al. 2024]. The $p=3.3$ line's ppl increases because it is within a range that our algorithm cannot converge.
>
>
> > 7. xi) Missing results for Stochastic ℓp Descent.
>
> Stochastic lp descent is a special case of our algorithm; thus we have presented the better empirical versions of our algorithm (with acceleration), though we will include ablation studies in the updated manuscript.
>
> > 8. (Q4) Stacey(p,2) vs. Stacey(p,p)?
>
> Basing our intuition on the theory for the convex case, coupling a non-Eulidean primal with a Euclidean dual update (Theorem 1 in [Bai & Bullins 2024]), vs. non-Euclidean primal and dual updates (e.g. Theorem 2 in [Diakonikolas & Guzmán 2024]), leads to different trade offs between the problem geometry, initial iterate, and acceleration exponent, such that neither is uniformly better than the other.
>
> > 9. (Q6) Stacey for p=2?
>
> This occurs due to numerical differences from handling general $p$. However, we acknowledge this could cause confusion, so we will provide the algorithm for the simplified case of Stacey(2,2), along with a discussion of this distinction.
>
> > **Choice of $p$**
>
> > (Q2) automatically work out "p"?
>
> One idea would be to use occasional Hessian information to adjust $p$ over time, based on the spectrum density [Ghorbani et al. 2019]. There has also been much work on developing automated means of hyperparameter tuning, including e.g. parameter-free methods [Jacobson & Cutkosky 2022], from which we hope to draw inspiration.
>
> > (Q3) adjusting Stacey p choice per layer?
>
> This is a wonderful suggestion. We may even benefit from a per-layer basis if we wished to leverage layer-wise Hessian information as a diagnostic tool (being far more compute-friendly).
>
> > **Theoretical Properties of Stacey**
>
> The acceleration framework of Stacey builds on linear coupling [Allen-Zhu & Orecchia, 2017], which is optimal in the first-order smooth and convex setting, and HASD [Bai & Bullins, 2024], which achieves faster convergence in the $\ell_p$ smooth non-Euclidean setting. Beyond the deterministic and convex regime, the core foundation of Stacey ---- non-convex stochastic $\ell_p$ steepest descent ---- achieves a first-order oracle complexity of $\mathcal{O}(\epsilon^{-4})$, which we discuss as tight in Section 4.1 and 4.2.
>
> Providing a further theoretical characterization of Stacey’s acceleration remains challenging. Notably, even for widely used algorithms like Adam, existing theory typically shows only convergence or recovers first-order optimal regret, with limited formal evidence of superiority over other accelerated first-order methods. We suspect capturing the acceleration of Stacey in theory would require more refined and tailored assumptions. While we leave this as an open direction for future work, we highlight that Stacey generalizes both SignSGD and Lion ---- as discussed in the third paragraph of Section 4.2 ---- which suggests it offers greater flexibility and is more likely to admit improved theoretical properties.

---

### Official Review · Reviewer_NAH2 · 2025-03-19

**Overall Recommendation:** 4

**Summary:**

This paper introduces **STACEY**, a novel optimization algorithm designed to accelerate stochastic steepest descent via ℓp-smooth nonconvex optimization. The key contributions of this work include:
- The development of **STACEY**, which incorporates primal-dual iterate interpolation to improve convergence rates for non-Euclidean smooth optimization problems.
- A theoretical framework that generalizes both SGD (when \( p = 2 \)) and signSGD (when \( p = \infty \)), with a **convergence guarantee of \( O(\epsilon^{-4}) \)** under standard assumptions.
- Empirical results demonstrating **superior convergence speed and final accuracy** compared to existing optimization methods, including SGD, Adam, AdamW, and Lion, on large-scale deep learning tasks such as image classification (CIFAR, ImageNet) and large language model (LLM) pretraining.
- A study on how different values of \( p \) affect performance, showing that non-Euclidean norms can be more effective in certain settings than traditional ℓ2-based methods.

The paper is **well-written, well-structured, and presents both strong theoretical contributions and compelling empirical results**.

**Claims And Evidence:**

The paper makes several claims, all of which are generally well-supported:
- **Claim:** STACEY improves convergence rates over traditional SGD and adaptive optimizers.
  **Evidence:** Empirical evaluations on CIFAR, ImageNet, and LLM pretraining demonstrate improved speed and final accuracy.
- **Claim:** The proposed method generalizes previous optimization approaches by considering a broader class of ℓp-norms.
  **Evidence:** Theoretical analysis rigorously proves this generalization.
- **Claim:** STACEY benefits from the flexibility of choosing different \( p \) values.
  **Evidence:** Experiments with different values of \( p \) show that problem-specific norm choices can yield better performance.

One area where **the claims could be strengthened** is in providing a more detailed computational complexity comparison to confirm the practical efficiency of STACEY in large-scale settings.

---

**Essential References Not Discussed:**

The paper **covers most essential references**, but **a comparison with curvature-aware optimizers (e.g., Shampoo, K-FAC)** would be useful. These methods also attempt to handle **non-Euclidean optimization challenges**, making them relevant to the discussion.

I recommend citing **Gupta et al. (2018) on Shampoo** and **Martens & Grosse (2015) on K-FAC** to highlight how STACEY differs from these approaches.

---

**Experimental Designs Or Analyses:**

The **experimental design is well-structured**, with meaningful comparisons. I verified the validity of:
- **Image classification experiments** on CIFAR and ImageNet.
- **LLM pretraining experiments** on the C4 dataset.
- **Ablation studies** exploring the role of \( p \).

Potential improvement:
- A **detailed analysis of computational efficiency (runtime per iteration, memory overhead, etc.)** would make the comparisons more complete.
- It would be useful to explore **STACEY’s performance in additional non-Euclidean problems**, such as adversarial training or reinforcement learning.

---

**Methods And Evaluation Criteria:**

The methods and evaluation criteria are **appropriate and well-justified**:
- **Optimization Benchmarks:** The paper evaluates STACEY on well-established benchmarks, including CIFAR-10, ImageNet, and LLM pretraining tasks.
- **Comparisons:** The algorithm is compared against widely used optimizers (SGD, Adam, AdamW, Lion), which are the **correct baselines** for this type of work.
- **Evaluation Metrics:** The paper reports **training loss, test accuracy, and convergence speed**, which are standard and relevant evaluation criteria for optimization algorithms in deep learning.

However, **an additional runtime or computational cost comparison** would be useful to fully assess the trade-offs of using STACEY in practice.

---

**Other Comments Or Suggestions:**

- **Clarify the intuition behind acceleration for different values of \( p \)**.
- **Include an ablation study on the effect of different values of \( p \) on generalization performance**.
- **Discuss whether STACEY could be extended to second-order methods or mixed-order approaches**.

---

**Other Strengths And Weaknesses:**

### **Strengths**
- **Strong theoretical foundation** with **generalized convergence guarantees**.
- **Comprehensive empirical validation** showing consistent improvements over baselines.
- **Clear writing and structured explanations**.

### **Weaknesses**
- **No computational cost analysis** (memory and runtime comparisons are missing).
- **Hyperparameter selection for \( p \) is unclear** (no systematic guidance provided).
- **Missing comparisons with curvature-aware optimizers** (e.g., Shampoo, K-FAC).

---

**Questions For Authors:**

1. **How does the per-iteration computational cost of STACEY compare to SGD, Adam, and Lion in terms of runtime and memory consumption?**
2. **Could you provide a practical heuristic or automated procedure for selecting \( p \) in different tasks?**
3. **How does STACEY perform when compared to second-order methods like Shampoo or K-FAC? Would these methods benefit from a similar ℓp-based approach?**

---

**Relation To Broader Scientific Literature:**

This paper is **well-grounded in prior research** on stochastic optimization and non-Euclidean geometry in machine learning:
- It builds upon **SGD (Robbins & Monro, 1951), signSGD (Bernstein et al., 2018), and AdamW (Loshchilov & Hutter, 2019)**.
- It connects with recent studies on **non-Euclidean optimization (Diakonikolas & Guzmán, 2024)** and **adaptive gradient methods**.
- The idea of **primal-dual interpolation** is influenced by **work on non-Euclidean acceleration (Allen-Zhu & Orecchia, 2017; Nemirovskii & Nesterov, 1985)**.

This paper **extends these ideas in a meaningful way**, demonstrating both theoretical and empirical improvements.

---

**Theoretical Claims:**

I reviewed the theoretical claims and found them **mostly sound**:
- The **convergence proof for stochastic ℓp steepest descent** follows standard assumptions and **logically extends previous results**.
- The **use of primal-dual interpolation** is well-motivated, and the acceleration rate follows existing literature on non-Euclidean acceleration.
- The **generalization to different ℓp norms** appears correct and aligns with prior research on optimization under non-Euclidean norms.

One possible **area for clarification** is the effect of different values of \( p \) on the acceleration exponent. **An intuitive explanation of how acceleration scales with \( p \) would improve clarity.**

---

---

> ### Author Rebuttal · Authors · 2025-04-01
>
> We thank the reviewer for their thoughtful comments and suggestions, and we respond to their questions below.
>
> > Detailed **computational complexity in terms of runtime and memory consumption** compared to SGD, Adam, and Lion
>
> **Runtime**
>
> Let $d$ be the number of parameters, and let each “basic operation” refer to simple scalar arithmetic (e.g., an addition, multiplication, or sign). We will ignore lower-level details (e.g., hardware vectorization) and focus on how many scalar operations are performed per parameter per iteration.
>
> **SGD**
>
> Key steps:
>
> 1) $m_i \leftarrow \beta m_i + (1-\beta)\nabla_i \quad (\text{2 multiplications, 1 addition})$.
> 2) $x_i \leftarrow x_i - \alpha m_i \quad (\text{1 multiplication, 1 addition})$.
>
> Approximate ops per parameter: ~5–6 scalar ops (fewer if no momentum is used).
>
> **Adam**
>
> Key steps:
>
> 1) $m_i \leftarrow \beta_1 m_i + (1-\beta_1)\nabla_i
> \quad (\text{2 multiplications, 1 addition})$.
> 2) $v_i \leftarrow \beta_2 v_i + (1-\beta_2)\nabla_i^2
> \quad (\text{3 multiplications, 1 addition})$.
> 3) $x_i \leftarrow x_i - \alpha\frac{m_i}{\sqrt{v_i} + \epsilon}
> \quad (\text{1 square root, 1 division 1 multiplication, 1 addition})$.
>
> Approximate operations per parameter: ~9–12 scalar ops (slightly more if you count the bias-correction steps separately).
>
> **Lion**
>
> Key steps:
>
> 1) $m_i \leftarrow \beta m_i + (1-\beta)\mathrm{sign}(\nabla_i)
> \quad (\text{2 multiplications, 1 addition, 1 sign operation})$.
> 2) $x_i \leftarrow x_i - \alpha\,\mathrm{sign}(m_i) \quad (\text{1 sign, 1 multiplication, 1 addition})$.
>
> Approximate operations per parameter: ~6–7 scalar ops (including sign as an operation).
>
> **Stacey**
>
> Key steps:
>
> 1) Update the “momentum-like” buffer $m_i$ (coordinate-wise re-scaling).
> 2) Update the dual vector $z_i$ (coordinate-wise multiplications/additions).
> 3) Combine $m_i$ and $z_i$ to get the final parameter update.
>
> Representative breakdown:
> 1) Update $m_i$: ~3–5 ops.
> 2) Update $z_i$: ~3–4 ops (coordinate-wise re-scaling plus addition).
> 3) Final parameter update: ~2–3 ops (linear combination and one addition/subtraction).
>
> Approximate operations per parameter: ~9–12 scalar ops.
>
> **Memory Footprint**
>
> **SGD**:Momentum buffer (if used). Total auxiliary overhead: $d$
>
> **Adam**: First moment and second moment. Total auxiliary overhead: $2d$
>
> **Lion**: Momentum-like buffer. Total auxiliary overhead: $d$
>
> **Stacey**: Momentum-like buffer and dual vector. Total auxiliary overhead: $2d$
>
> Thus, Stacey’s memory requirement is similar to Adam, and slightly more than other single-pass gradient methods (though we note that its overhead compared to e.g. SGD, Lion comes precisely from the additional dual vector).
>
> > **Choice of $p$: intuition, ablation, practical heuristics**
>
> Although it can be challenging to provide intuition for acceleration (even in the convex $p=2$, i.e. Nesterov's, case), at a high level the analysis carefully balances the primal and dual updates, leveraging the uniform convexity of $\|\cdot\|_p^p$ and an alternative smoothness-derived upper bound, as in Definition 2 and Lemma 1 in [Diakonikolas & Guzmán 2024]. We will include an ablation study of the influence of $p$ on the acceleration rate in the updated manuscript, to help provide additional insight for these theoretical results.
>
> Regarding heuristics for choosing $p$, one idea would be to use occasional Hessian information to determine how to adjust $p$ over time, based on the spectrum density [Ghorbani et al. 2019]. Additionally, there has been much work on developing automated means of hyperparameter tuning, including e.g. parameter-free methods [Jacobson & Cutkosky 2022], from which we may hope to draw inspiration.
>
> > **Second-order curvature-aware optimizers: citations and discussion**
>
> We thank the reviewer for pointing out these helpful references. We will cite them and incorporate the following discussion in the revised version. Shampoo [Gupta et. al., 2018], K-FAC [Martens & Grosse, 2015] and their follow-ups are indeed representative works of curvature-aware optimization methods. One notable difference is that these works are second-order methods that exploit the structure of the Hessian or Fisher information and focus on techniques for their efficient approximation, whereas our method, Stacey, is a first-order approach that explores non-Euclidean geometry though a differing $\ell_p$ norm. We will include comparisons with these curvature-aware optimizers in the updated version of our manuscript.
>
> Extending Stacey to second-order methods is, in our view, a promising direction that aligns with our own considerations for future research as well. It is natural to investigate the non-Euclidean counterpart of a preconditioned gradient step, just as we have done in the first-order setting. Such a method holds the potential to benefit from both the curvature awareness provided by the Hessian information and the geometric advantages of operating under a differing $\ell_p$ norm.

---

### Decision · Program_Chairs · 2025-05-01

**Decision:**

Accept (poster)

**Comment:**

This paper introduces Stacey, an accelerated l_p steepest descent algorithm leveraging primal-dual interpolation for non-Euclidean optimization. Theoretical guarantees and empirical evaluations on image classification and LLM pretraining suggest improved convergence speed, final accuracy, and benefits from adaptive p-values, advocating for non-Euclidean methods over conventional approaches.

The reviews are highly mixed: Reviewers d7Gi  and Reviwer  NAH2 strongly support the work, emphasizing its novel theoretical insights into optimizer design, which could inspire future research. However, Reviewer ddGc raises significant concerns. The primary criticism centers on the experimental validity, arguing that Stacey’s performance may not surpass Adam, let alone state-of-the-art optimizers (e.g., Shampoo, Muon, Scion). The AC also notes Stacey’s hyperparameter complexity, which may limit practical adoption.

While the empirical claims require further validation, the theoretical contributions—particularly the framework for non-Euclidean optimization—align with ICML’s theory-oriented focus. Balancing these factors, the AC leans toward acceptance, as the work provides  inspirations for future research into simpler, effective optimizers for realistic deep learning scenarios.